# Sonogenetic control of mammalian cells using exogenous Transient Receptor Potential A1 channels

Marc Duque [1,4], Corinne A. Lee-Kubli[1,4], Yusuf Tufail[1,4], Uri Magaram[1,2], Janki Patel [1], Ahana Chakraborty [1], Jose Mendoza Lopez [1], Eric Edsinger[1], Aditya Vasan[3], Rani Shiao[1], Connor Weiss[1], James Friend[3] & Sreekanth H. Chalasani [1,2✉]

Ultrasound has been used to non-invasively manipulate neuronal functions in humans and other animals. However, this approach is limited as it has been challenging to target specific cells within the brain or body. Here, we identify human Transient Receptor Potential A1 (*hs*TRPA1) as a candidate that confers ultrasound sensitivity to mammalian cells. Ultrasound-evoked gating of *hs*TRPA1 specifically requires its N-terminal tip region and cholesterol interactions; and target cells with an intact actin cytoskeleton, revealing elements of the sonogenetic mechanism. Next, we use calcium imaging and electrophysiology to show that *hs*TRPA1 potentiates ultrasound-evoked responses in primary neurons. Furthermore, unilateral expression of *hs*TRPA1 in mouse layer V motor cortical neurons leads to *c-fos* expression and contralateral limb responses in response to ultrasound delivered through an intact skull. Collectively, we demonstrate that *hs*TRPA1-based sonogenetics can effectively manipulate neurons within the intact mammalian brain, a method that could be used across species.

[1] Molecular Neurobiology Laboratory, The Salk Institute for Biological Studies, La Jolla, CA 92037, USA. [2] Neurosciences Graduate Program, University of California San Diego, La Jolla, CA 92093, USA. [3] Medically Advanced Devices Laboratory, Department of Mechanical and Aerospace Engineering, Jacobs School of Engineering, University of California San Diego, La Jolla, CA 92093, USA. [4] These authors contributed equally: Marc Duque, Corinne A. Lee-Kubli, Yusuf Tufail. ✉email: schalasani@salk.edu

Ultrasound is safe, noninvasive, and can be easily focused through thin bone and tissue to volumes of a few cubic millimeters[1,2]. Moreover, continuous or repeated pulses of ultrasound at fundamental frequencies between 250 kHz and 3 MHz have been shown to stimulate neurons in rodents and nonhuman primates[3–7]. Ultrasound has also been used to safely manipulate deep nerve structures in human hands to relieve chronic pain[8], as well as to elicit somatosensory[9] and visual cortex sensations[10] through the intact skull. These and other studies have revealed a wide interest in adapting ultrasound for both research and therapeutic purposes[11]. Nevertheless, the mechanisms that underlie ultrasound neurostimulation remain unclear, but may include mechanical forces[12], heating[8], cavitation[11], and astrocyte signaling[13] in vitro, or indirect auditory signaling within the rodent brain in vivo[14,15]. We and others have found evidence for the involvement of mechanosensitive channels in ultrasound responses of naïve rodent neurons in vitro[16] and C. elegans neurons in vivo[17,18]. This provides a potential path toward the development of a broadly usable sonogenetic tool that would target exogenous proteins to specific cells, thereby rendering them sensitive to ultrasound stimuli at pressures and durations that do not affect naïve cells.

We previously showed that exogenous expression of the C. elegans TRP-4 mechanoreceptor enables ultrasound sensitivity in neurons that are otherwise unresponsive to ultrasound stimulation[18]. Similar ultrasound sensitivity has also been observed in vitro in cells induced to express proteins belonging to the mechanosensitive (MSC)[19], Piezo[20], Prestin[21], transient receptor potential (TRP)[16], and TREK[22] families. However, the frequencies and pressures used in most of these studies (500kHz-2MHz, 10 MHz was used to show activation of TREK channels in Xenopus oocytes[22]) overlap with those reported to induce endogenous responses in neurons, have limited spatial resolution, or require bulky transducers, all of which restrict the ability to develop wearable devices. In addition, while lower frequency stimuli are more likely to penetrate biological tissue, they have large focal areas making it difficult to target specific brain regions (Fig. 1a). To overcome these challenges, we use ultrasound at 7 MHz, which can be focused to a small volume of 107 μm³, suitable for applications in the rodent model. Moreover, this frequency is unlikely to induce cavitation as the mechanical index range used in our experiments (0.37–0.95) is well below the threshold for cavitation of 1.9 in tissues[23,24].

We therefore set out to find new proteins from ion channel families thought to have mechanosensitive properties, which could confer ultrasound sensitivity to mammalian cells at higher frequencies (7 MHz). To identify an optimal candidate, we use a functional readout-based assay to screen a library of 191 candidate channels and their homologs (Supplementary Table S1). We then combine imaging, pharmacology, electrophysiology, and comparative sequence analysis, as well as behavioral, and histological analyses to demonstrate that a mammalian protein, Homo sapiens transient receptor potential A1 (hsTRPA1), confers ultrasound sensitivity to cells in vitro and in vivo, thereby establishing a sonogenetic tool in mammals.

## Results

**hsTRPA1 is a sonogenetic candidate**. We first estimated the focal volume of ultrasound at different frequencies along with their penetration success in biological tissue using standard equations (see methods for details). These calculations show that while lower frequencies can penetrate 5 mm of brain tissue with minimal loss in energy, they deliver energy to an mm³-sized volume. Conversely, higher frequency ultrasound (> 10 mHz) can be focused to a small volume, but is unable to penetrate brain tissue. We chose 7 MHz as an optimal frequency as it can be focused to 107 μm³, suitable for applications in the rodent model (Fig. 1a). Next, we aligned an imaging setup with a custom-designed, single-crystal 7 MHz lithium niobate transducer (Fig. 1b) that does not exhibit hysteresis and thereby generates minimal heat as it converts electrical input into mechanical energy[25]. We also profiled the pressure output and corresponding temperature changes in our imaging set up using a combined fiber optic probe (Supplementary Fig. S1a, b) and selected ultrasound parameters for the screen at a pressure and duration that caused minimal temperature change (100 ms, 1.5 MPa measured above the coverslip). The glass coverslip attenuated peak negative pressure by 75% relative to the pressure output of the transducer. We found that propidium iodide was unable to penetrate the cells exposed to these ultrasound stimuli at 2.5 MPa or below, confirming that these parameters do not disrupt cellular membranes (Supplementary Fig. S1d–m). We transiently transfected each of 191 candidate proteins (Supplementary Table S1) along with a dTomato (dTom) fluorescent reporter into human embryonic kidney-293T (HEK) cells expressing a genetically encoded calcium indicator (GCaMP6f[26], Supplementary Fig. S2a, b) and monitored changes in intracellular calcium upon ultrasound stimulation (Supplementary Fig. S2c, d, f, g). We found that cells expressing mammalian TRPA1 channels responded to ultrasound most frequently, with the human homolog as the most effective candidate (Fig. 1c and Supplementary Video S1). Moreover, while we observed that most dTom control cells do not respond to ultrasound stimuli, a significant fraction of hsTRPA1-expressing cells have robust responses (Supplementary Fig. S1c and 2d). In contrast, the mouse homolog was only a third as responsive as hsTRPA1, and nonmammalian variants were insensitive to ultrasound (Fig. 1d), while still showing comparable responses to a chemical agonist, allyl isothiocyanate (AITC) (Supplementary Fig. S2l). None of the other proteins tested showed significant sensitivity to ultrasound parameters used in our screen, including channels previously shown to respond to ultrasound stimuli at different frequencies, pressures, or durations (Fig. 1d). While we confirmed functional overexpression of Piezo1 and TRPV1 candidate channels (Supplementary Fig. S2j, k), we cannot rule out the possibility that other tested proteins did not perform due to issues with expression, trafficking, or folding. Since hsTRPA1-expressing cells showed a robust response to ultrasound stimuli at 2.5 MPa and 100 ms duration (Supplementary Fig. S1c), we used these parameters for the rest of our in vitro studies unless indicated otherwise. Next, we compared the ultrasound responsiveness of hsTRPA1 with that of previously identified sonogenetic candidates (MscL, Prestin, TRPV1 and Piezo1) at different frequencies. We found that hsTRPA1 was more effective than these other candidates at frequencies of 1 MHz, 2 MHz, and 7 MHz, confirming that hsTRPA1 channel was sensitive to a broad range of ultrasound stimuli (Fig. 1f).

We next confirmed that ultrasound responsiveness was due to direct activation of hsTRPA1. We first visualized hsTRPA1 using immunohistochemistry and found that it was indeed expressed only in dTom+ cells and trafficked to HEK plasma membranes (Fig. 2a), where it co-localized with membrane-targeted EGFP-CAAX[27]. However, we do observe that only a small fraction of hsTRPA1 is detected on the membrane and cannot rule out a role for this protein in other cellular compartments. We also consistently found that HEK cells expressing hsTRPA1 were selectively activated by ultrasound stimulation in a pressure- and duration-dependent manner (Supplementary Figs. S1c and S2d), while dTom-only control cells showed no response to ultrasound stimulation (Supplementary Fig. S1c and S2g and Supplementary Video S2). Moreover, we found that the TRPA1-selective agonists N-methylmaleimide (NMM) and AITC[28] also specifically

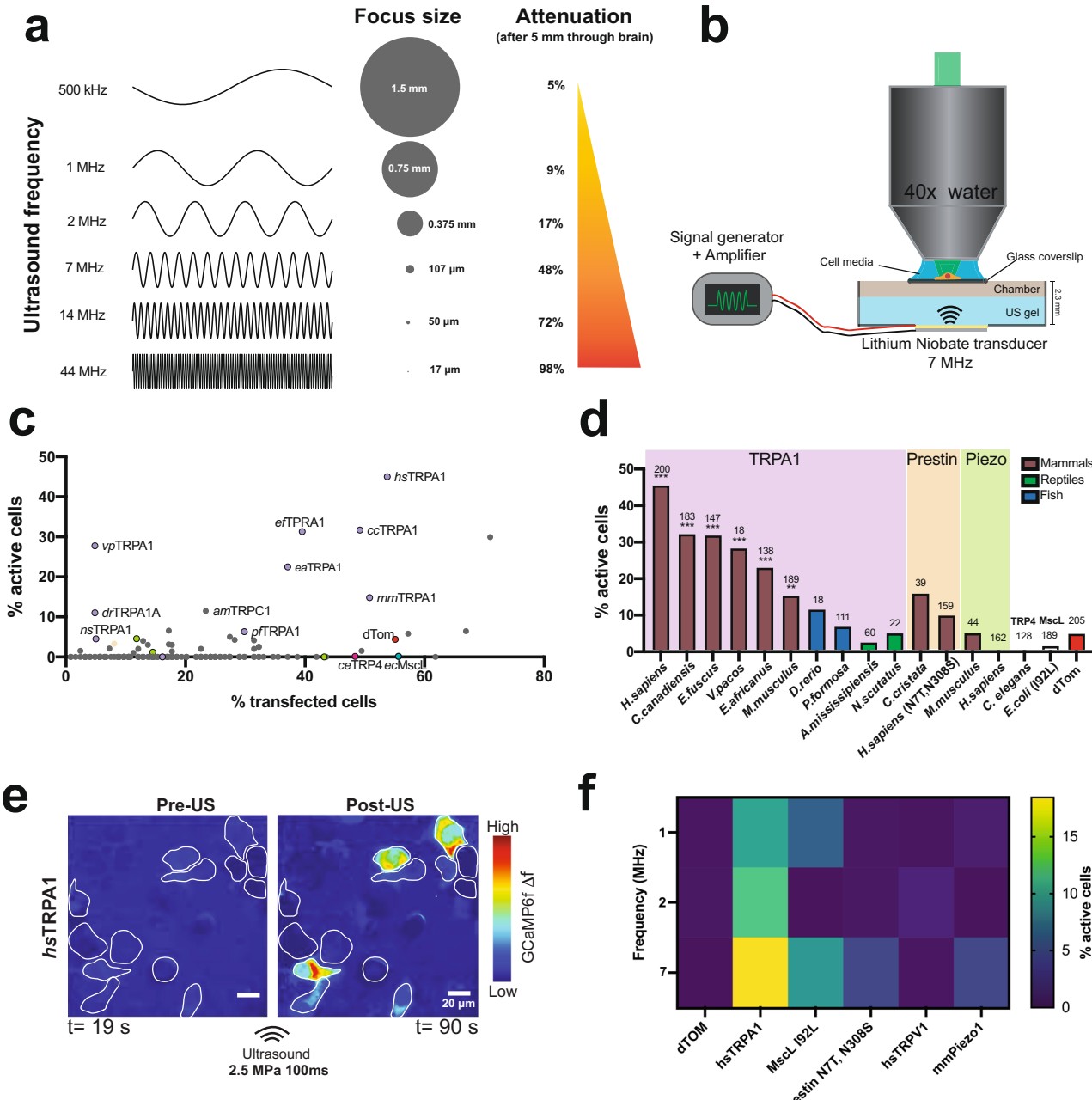

**Fig. 1 2D screen identifies *hs*TRPA1 as a candidate for sonogenetic stimulation at 7 MHz. a** Focal area and penetrance estimated for different ultrasound frequencies. **b** Schematic showing the 7 MHz lithium niobate transducer delivering ultrasound stimuli to cells. Plot showing **c**, the percent of transfected cells vs percent of transfected cells that were activated after ultrasound stimulation for 191 cDNAs and **d**, the top responders and their homologs compared to previously published ultrasound-sensitive candidates. For the screen, $N = 3$ coverslips/clone and each coverslip was imaged 3 times. **e** GCaMP6f signal in HEK cells expressing *hs*TRPA1 before and after ultrasound stimulation. ROIs identify transfected cells (dTom). Scale bar 20 µm. **f** Comparison of previously published ultrasound-sensitive candidates and *hs*TRPA1 in terms of % active cells after ultrasound stimulation at different frequencies (1 MHz, 2 MHz and 7 MHz). $N = 3$ coverslips/condition. The numbers of cells analyzed is indicated above each bar. **d** **$p < 0.01$, ***$p < 0.001$ by logistic regression. Source data are provided with this paper.

activated TRPA1-expressing HEK cells, confirming that the channel was indeed functional (Fig. 2b and Supplementary Fig. S2e, h, i). Additionally, the TRPA1 antagonist HC-030031[29] inhibited ultrasound responses in *hs*TRPA1-expressing cells (Fig. 2b). Collectively, these results show that the ultrasound responses require gating of *hs*TRPA1, which facilitates intracellular calcium increases.

Next, we used electrophysiological methods to monitor changes in the membrane conductance of excitable HEK cells

(exHEK)[30] expressing *hs*TRPA1 or dTom-only control. To increase the recording efficiency, we used a cell-attached configuration (Fig. 2c), wherein we were able to maintain suitable access and membrane resistance while exposing cells to ultrasound stimuli. We found that cells expressing *hs*TRPA1 had higher basal rates of activity (Fig. 2d) but no significant disruptions in their I-V curve compared to controls (Fig. 2e), confirming that channel expression did not alter membrane properties. Inward currents in response to ultrasound were

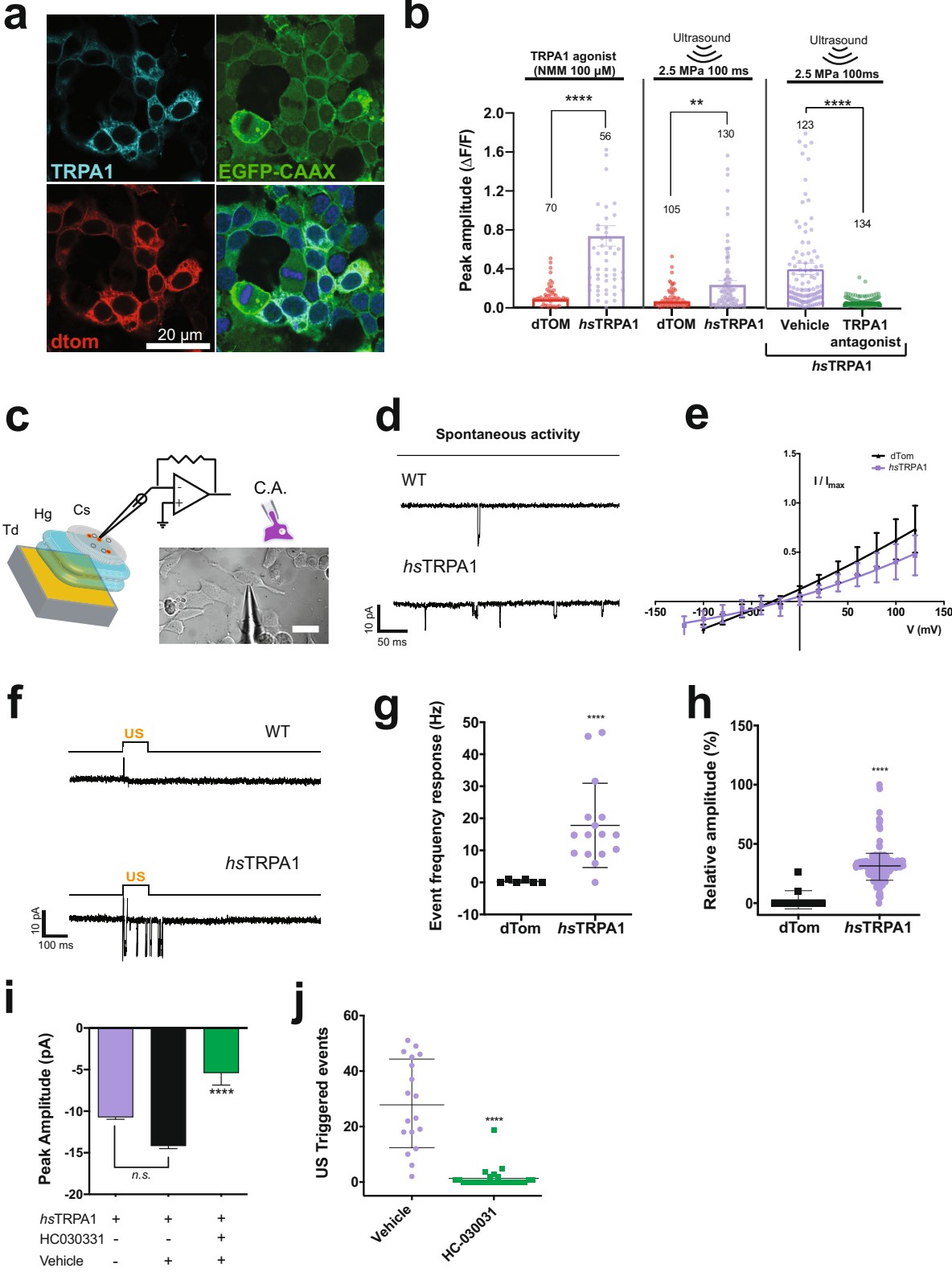

significantly larger and more frequent in *hs*TRPA1-expressing cells compared to controls (Fig. 2f–h). These ultrasound-triggered currents were of similar magnitude as those previously observed for pharmacological activation of the TRPA1 channel[31]. It is possible that exHEKs expressing *hs*TRPA1 channels could be activated without ultrasound stimulation in our cell-attached configuration, due to membrane stretch that is intrinsic to the procedure. However, our spontaneous gap-free recordings showed minimal currents, indicating that the slow and sustained membrane stretch, or the membrane seal itself, did not have a large effect comparable to what we observed with ultrasound stimulation. Also, we did not test the conductance of *hs*TRPA1 expressing cells in response to increasing vacuum pressure as we were focused on assessing the effects of ultrasound-mediated

**Fig. 2 Functional characterization of *hs*TRPA1 ultrasound responses in HEK cells. a** Representative image showing *hs*TRPA1 expression co-localized with membrane-targeted EGFP-CAAX in HEK cells. **b** GCaMP6f peak amplitude in *hs*TRPA1- or dTom–expressing (control) HEK cells stimulated with TRPA1 agonist (NMM, 100 µM), ultrasound alone or TRPA1 antagonist (HC-030031 40 µM.). **c** Schematic showing the cell-attached configuration for electrophysiology with a differential interference contrast (DIC) microscopy image of a representative HEK cell. Td: ultrasound transducer, Hg: ultrasound gel, Cs: coverslip. **d** Representative traces in HEK cells expressing dTom only (control) or *hs*TRPA1 before ultrasound stimulation, showing increased spontaneous activity in *hs*TRPA1-expressing cells. **e** I-V plot of HEK cells expressing dTom control or *hs*TRPA1. **f** Representative gap-free voltage-clamp trace of dTom control- or hsTRPA1-expressing HEK to 100 ms, 0.15 MPa ultrasound stimuli. **g** HEK cells expressing *hs*TRPA1 have more frequent ultrasound-triggered membrane events compared to dTom controls. **h** Summary of relative peak amplitude responses (I/Imax) in HEK cells expressing dTom or *hs*TRPA1. **i** Mean peak amplitude (pA) from HEK cells expressing *hs*TRPA1 alone, and *hs*TRPA1 treated with vehicle or TRPA1 antagonist (HC-030031 40 µM). **j** Individual and average number of events post ultrasound stimulus in *hs*TRPA1 expressing cells vs. *hs*TRPA1 + HC030031. Data are mean ± SEM (**b**, **e**, **g**, **h**, **i**, **j**). $n = 3$ (**b**) independent coverslips or $n = 8$ cells/group with at least 2 trials per cell (**e**, **g–j**). **p < 0.01, ****p < 0.0001 by two-tailed Mann-Whitney test (**b**). ****$p < 0.0001$ compared to control by unpaired, two-tailed *t* test with Welch's correction (**g–i**). Numbers of cells analyzed is indicated above each bar (**b**). Source data are provided with this paper.

stimulation. Furthermore, we found that both the amplitude and number of ultrasound-triggered membrane events were attenuated by the TRPA1 antagonist, HC-030031, confirming a specific role for *hs*TRPA1 (Fig. 2i, j). Taken together, these results show that ultrasound pulses can selectively lead to the opening of *hs*TRPA1 channels, resulting in a rapid increase of intracellular calcium in HEK cells.

**Putative mechanisms underlying ultrasound sensitivity of TRPA1.** Multiple studies have shown that TRPA1 is a widely conserved calcium-permeable nonselective cation channel that is involved in detecting a wide range of exogenous stimuli including electrophilic compounds that interact with the nucleophilic amino acids in the channel, small peptides that partition in the plasma membrane, cold, heat, and others, although sensitivity to different stimuli varies across species (reviewed by[32]). Despite this broad sensitivity and a resolved crystal structure, the underlying mechanisms of TRPA1 activation are only recently being elucidated. For example, a scorpion toxin peptide (WaTx) has been shown to activate TRPA1 by penetrating the lipid bilayer to access the same amino acids bound by electrophiles, thereby stabilizing the channel in an active state and prolonging channel opening[33]. In contrast, electrophilic irritants have been shown to activate the TRPA1 channel using a two-step cysteine modification that widens the selectivity filter to enhance calcium permeability and open the cytoplasmic gate[34]. These studies suggest that the TRPA1 channel might interact with the cytoskeleton and components of the membrane bilayers, including cholesterol, to transduce signals.

Structurally, TRPA1 comprises an intracellular N-terminal tip domain, 16 ankyrin repeats, 6 transmembrane domains, and an intracellular C-terminal domain (Supplementary Fig. S3a). To identify TRPA1 domains critical for ultrasound sensitivity, we compared sequences of each domain in the human protein to its ultrasound-sensitive mammalian and ultrasound-insensitive non-mammalian chordate TRPA1 homologs (Supplementary Fig. S3b and Supplementary Table S2). We expected that *hs*TRPA1 domains and motifs that are specifically conserved among mammals specifically may be crucial for ultrasound sensitivity. Sequence analysis of *hs*TRPA1 and the 9 additional homologs we tested revealed that the 61 amino acid N-terminal tip region is highly conserved in the mammalian compared to non-mammalian chordate species that we tested (58 vs. 13% identity, respectively), particularly the first 22 amino acids (87 vs. 17% identity, respectively, Fig. 3a). Therefore, we hypothesized that the N-terminal tip region might be important for mediating ultrasound sensitivity. Indeed, deletion of the entire N-terminal tip region (Δ1–61) and or the most highly conserved portion (Δ1–25) from *hs*TRPA1 completely abolished responses to ultrasound (Fig. 3b), while significantly increasing sensitivity to

chemical agonist (Fig. 3c). In contrast to the N-terminal tip region, the ankyrin repeat regions are highly conserved across both tested mammals (82% identity) and nonmammalian species (54% identity), with the exception of ankyrin 1, which is least conserved across mammals (46% identity; Supplementary Table S2 and Supplementary Fig. S3c). Therefore, we hypothesized that ankyrin 1 would not be required for the ultrasound response. Indeed, deletion of only the first ankyrin repeat (ΔANK1) had no effect on either sensitivity to ultrasound or the chemical agonist (Fig. 3b, c). In order, to further confirm the importance of the human N-terminal tip for mediating ultrasound sensitivity, we created chimeras consisting of the alligator or zebrafish N-terminal tip swapped into *hs*TRPA1. These chimeras completely lost the ability to respond to ultrasound (Fig. 3d). However, the alligator/*hs*TRPA1 chimera had attenuated responses to AITC (Fig. 3e), suggesting that this channel may also have altered functionality. In contrast, the zebrafish/*hs*TRPA1 chimera had normal responses to AITC (Fig. 3e). Nevertheless, immunohistochemistry showed comparable expression and trafficking of mutated channels, indicating their lack of ultrasound responses is not a consequence of poor expression (Supplementary Fig. S4). Taken together, these data suggest that the human N-terminal tip region is important for *hs*TRPA1 ultrasound sensitivity.

The N-terminal ankyrin repeats have also been hypothesized to interact with cytoskeletal elements and act as a gating spring in response to mechanical stimuli[35,36]. For example, ankyrin repeat regions from *Drosophila* NOMPC (TRPN) are thought to be important in mechanosensation due to their interactions with microtubules[37]. So, we probed the involvement of cytoskeletal elements in ultrasound sensitivity of *hs*TRPA1. We found that treating *hs*TRPA1-expressing HEK cells with the actin-depolymerizing agents, cytochalasin D and latrunculin A, reduced their ultrasound responses compared to vehicle or an actin stabilizing agent, jasplakinolide (Fig. 3f). In contrast, disrupting or stabilizing microtubules with nocodazole or Taxol, respectively, had no significant effect on ultrasound-evoked *hs*TRPA1 responses (Fig. 3f). Immunohistochemistry confirmed that destabilizing treatments did indeed disrupt the actin cytoskeleton and microtubules (Supplementary Fig. S5a, b). Moreover, we found that AITC-triggered responses were not altered by treatment with either cytochalasin D or nocodazole, although they were significantly reduced by latrunculin A, jasplakinolide, and paclitaxel, confirming that these treatments did not completely disrupt TRPA1 function (Supplementary Fig. S5c). While latrunculin and cytochalasin D are both actin-depolymerizing agents; they use distinct mechanisms to disrupt actin filaments[38], which could explain why they have different effects on channel function and chemical agonist responses. Collectively, we found that actin depolymerization by

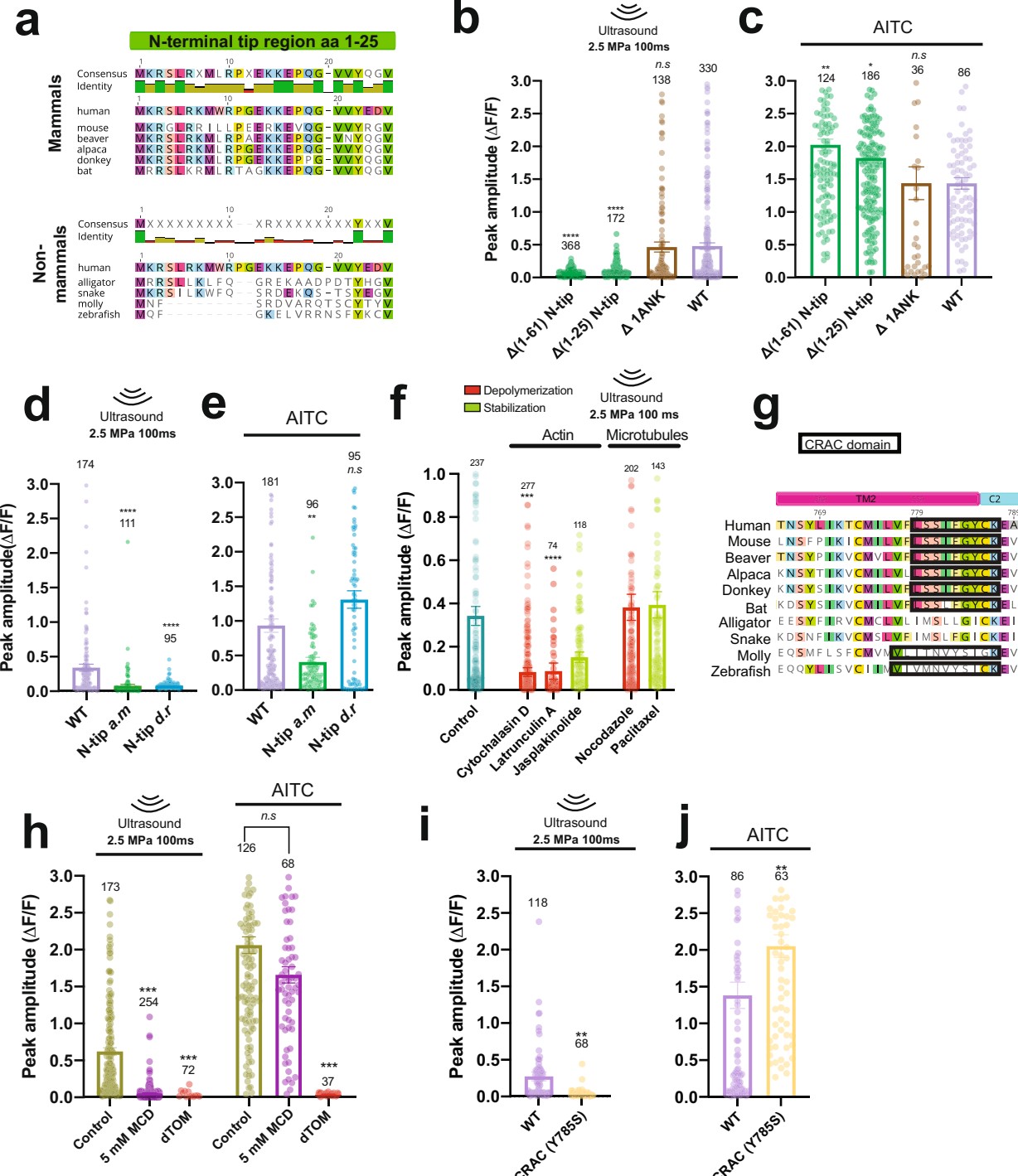

**Fig. 3 The N-terminal region of hsTRPA1, actin cytoskeleton, and cholesterol contribute to ultrasound sensitivity. a** Mammalian and non-mammalian alignments of the TRPA1 N-terminal tip region (aa1–25) from homologs tested for ultrasound sensitivity. GCaMP6f peak amplitude upon **b**, ultrasound stimulation or **c**, treatment with AITC (33 μM) in HEK cells transfected with either full-length *hs*TRPA1 or channels containing deletions of the whole N-terminal tip (Δ(1–61)), an initial subsection of the N-tip (Δ(1–25)) or only ankyrin repeat 1 (ΔANK1) without altering the pore or transmembrane regions. GCaMP6f peak amplitude upon **d**, ultrasound stimulation or **e**, treatment with AITC (33 μM) in HEK cells transfected with either full-length *hs*TRPA1 or chimeras in which the N-tip from alligator TRPA1 (N-tip a.m) or from zebrafish (N-tip d.r) was swapped in. **f** GCaMP6f peak amplitude following ultrasound stimulation in cells expressing *hs*TRPA1 after treatment with agents that either stabilize (green) or destabilize (red) microtubules and actin filaments compared to vehicle control. **g** Transmembrane 2 domain sequence alignment across species tested for ultrasound sensitivity with Cholesterol Recognition/interaction Amino acid Consensus (CRAC) domain outlined. **h** GCaMP6f peak amplitude in *hs*TRPA1-expressing HEK cells upon ultrasound stimulation or AITC treatment (33 μM) after incubation with MCD (5 mM) or control. **i** GCaMP6f peak amplitude in HEK cells expressing either WT *hs*TRPA1 or a mutant with TM2 CRAC domain disrupted (Y785S) upon ultrasound stimulation or **j** AITC treatment (33 μM); Data are mean ± SEM (**b–f**, **h–j**); n = 3 (**b–f**, **h–j**) biologically independent replicates. Numbers on each bar indicate the numbers of cell analyzed. n.s p > 0.05, **p < 0.01, ***p < 0.001, ****p < 0.0001 by Kruskal-Wallis rank test and Dunn's test for multiple comparisons. p value = 0.0016 (**i**), p value = 0.0066 (**j**). Source data are provided with this paper.

cytochalasin D treatment selectively blocked *hs*TRPA1 responses to ultrasound, but not chemical agonist, demonstrating a specific role for the actin cytoskeleton in ultrasound sensation.

Recent reports have suggested that ultrasound stimulation generates free radicals and reactive oxygen species, which can in turn affect cellular responses[39]. Moreover, other studies have shown that TRPA1 can be activated by reactive oxygen species (ROS)[40,41]. To test whether ultrasound responsiveness of *hs*TRPA1 expressing cells requires ROS, we used a pharmacological approach. We found that treating cells with a ROS inhibitor, edavarone, had no effect on ultrasound-evoked responses in *hs*TRPA1 expressing cells (Supplementary Fig. S5d). These data suggest that ROS might not play a role in *hs*TRPA-1 triggered sonogenetic control in vitro.

Mouse TRPA1 has also been hypothesized to localize to lipid rafts through a mechanism governed by a cholesterol recognition/ interaction amino acid consensus sequence (CRAC) domain within the transmembrane helix 2 (TM2) of TRPA1[42]. Interestingly, we identified a CRAC motif (L/V-(X)(1–5)-Y-(X)(1–5)-R/ K, where X are nonpolar residues) in transmembrane helix 2 that was highly similar in all mammalian homologs tested, but was absent in reptiles and heavily modified in fish (Fig. 3g). Therefore, we hypothesized that interactions with cholesterol might be important for ultrasound responsiveness of *hs*TRPA1 channels. We depleted cellular membrane cholesterol with methyl-β-cyclodextrin (MCD, 5 mM)[43], and found that this treatment attenuated *hs*TRPA1 responses to ultrasound, but did not affect sensitivity to AITC (Fig. 3h and Supplementary Fig. S5e). In order to further explore the importance of the TM2 CRAC motif, we created a mutant channel in which the required central tyrosine was replaced with a serine. This mutation caused a complete loss of ultrasound sensitivity without decreasing responsiveness to AITC (Fig. 3i, j), confirming a role for this TM2 CRAC motif in *hs*TRPA1 gating by ultrasound. While the precise molecular mechanism of TRPA1 responsiveness to ultrasound remains elusive, our studies suggest a role for the N-terminal tip region, the actin cytoskeleton, and interaction with cholesterol in driving ultrasound-evoked *hs*TRPA1 responses.

**hsTRPA1 potentiates ultrasound-responses in primary neurons.** To test whether *hs*TRPA1 can also render neurons sensitive to ultrasound stimuli, we infected embryonic mouse primary cortical neurons with adeno-associated viral (AAV) vectors at day-in-vitro five expressing either Cre-dependent *hs*TRPA1 or Cre-only control along with GCaMP6f[26] (Fig. 4a). Consistent with previous studies[44], we did not detect *hs*TRPA1 RNA in dorsal root ganglia (DRG) or brains from E18 mice (Supplementary Fig. S6e, f). Next, we confirmed functional expression of *hs*TRPA1 in infected neurons by monitoring their responses to AITC, and likewise observed that Cre-only control neurons did not respond to AITC (Supplementary Fig. S7a). Consistent with our HEK cell results, we found that ultrasound triggered a significant increase in intracellular calcium in *hs*TRPA1-expressing neurons (Fig. 4b, c and Supplementary Video S3). Cre-expressing neurons showed some calcium responses to ultrasound, but these were significantly lower in magnitude than those observed in *hs*TRPA1-expressing neurons (Supplementary Fig. S7b, c, Supplementary Video S4 and Supplementary Fig. S8a–c). Both *hs*TRPA1 and control Cre-expressing neurons showed increased responses to longer (Supplementary Fig. S7d) and more intense ultrasound stimuli (Fig. 4d). These results are consistent with previous studies showing that ultrasound exposure does result in a small change in intracellular calcium in naïve neurons[16]. However, *hs*TRPA1-expressing neurons showed greater sensitivity and reduced response latency to ultrasound stimuli

(Supplementary Fig. S8d, e). The majority of *hs*TRPA1-expressing neurons had a response latency within 500–900 ms of stimulus onset, while response durations ranged from 2–30 s (Supplementary Fig. S8f, g). Moreover, *hs*TRPA1-expressing neurons could be repeatedly stimulated without apparent deleterious effects on cell health or a substantial decrement in calcium flux (Fig. 4e and Supplementary Video S5), with cells returning to baseline after stimulation.

Next, we probed whether ultrasound-mediated effects in control and *hs*TRPA1-expressing neurons were due to TRPA1. Treating *hs*TRPA1 expressing neurons with HC-030031 significantly attenuated their responses to ultrasound but had no effect on the ultrasound-evoked activity in control neurons (Supplementary Fig. S7e). Furthermore, cortical neurons cultured from TRPA1-/- mice also responded to ultrasound (Supplementary Fig. S7f), indicating that even undetectable levels of TRPA1 in neurons or astrocytes likely do not account for ultrasound responses in control neurons. Moreover, the sodium channel blocker, tetrodotoxin, partially blocked ultrasound responses in *hs*TRPA1 neurons, while completely abolishing responses in control neurons (Supplementary Fig. S7e). We also found that sequestering extracellular calcium with BAPTA blocked neuronal responses to ultrasound (Supplementary Fig. S8h). However, treating neurons with a TRPV1 antagonist had no effect on their ultrasound responses (Supplementary Fig. S8h), thereby ruling out the TRPV1 heat-responsive channel's contribution to ultrasound sensitivity in TRPA1-expressing neurons either directly or through a synergistic interaction[45]. These results are consistent with a recent study that identified multiple mechanosensitive channels that can transduce calcium responses to ultrasound in control neurons[16]. These results show that ultrasound can directly activate AAV9-*hs*TRPA1 transduced neurons, leading to intracellular calcium influx, which may be amplified by voltage-gated sodium channels. In contrast, ultrasound responses in control neurons are due to a TRPA1-independent mechanism.

We next used whole-cell electrophysiological methods to confirm the role of *hs*TRPA1 in mediating ultrasound-evoked neuronal responses. We were able to obtain stable membrane resistances and reliable measurements during ultrasound stimulation trials using a whole-cell patch-clamp configuration (Fig. 4f). However, we were only able to assay responsiveness to pressures below 0.5 MPa in order to ensure the integrity of the patch. Similar to our HEK cell experiments, we found that AAV-mediated *hs*TRPA1 expression did not alter rat primary neuronal membrane properties (Supplementary Fig. S9a). In voltage clamp, Cre-expressing control neurons showed inward currents in response to ultrasound, consistent with the previous studies[4,16]. However, *hs*TRPA1-expressing neurons showed larger current responses (>400 pA) than controls in response to ultrasound stimulation (Fig. 4g–i). Unlike our imaging experiments, we were unable to perform repeated ultrasound stimulations on the same neuron as the integrity of the membrane patch rapidly deteriorated when the ultrasound inter-stimulus interval was <30 s. Taken together, hsTRPA1-expressing neurons had enhanced responses to ultrasound relative to control as assessed by their relative response, the magnitude of peak responses, and area under the curve metrics (Supplementary Fig. S9b–f). We further evaluated responsiveness to ultrasound in current-clamp mode to evaluate action potential generation. Ultrasound stimulation successfully triggered action potentials in neurons expressing *hs*TRPA1 (Fig. 4j) at measured pressures of 0.5 MPa, whereas control neurons showed subthreshold changes in membrane voltage that were insufficient to trigger action potentials during a majority of stimulation trials (Fig. 4k). Collectively, all assayed *hs*TRPA1-expressing neurons showed

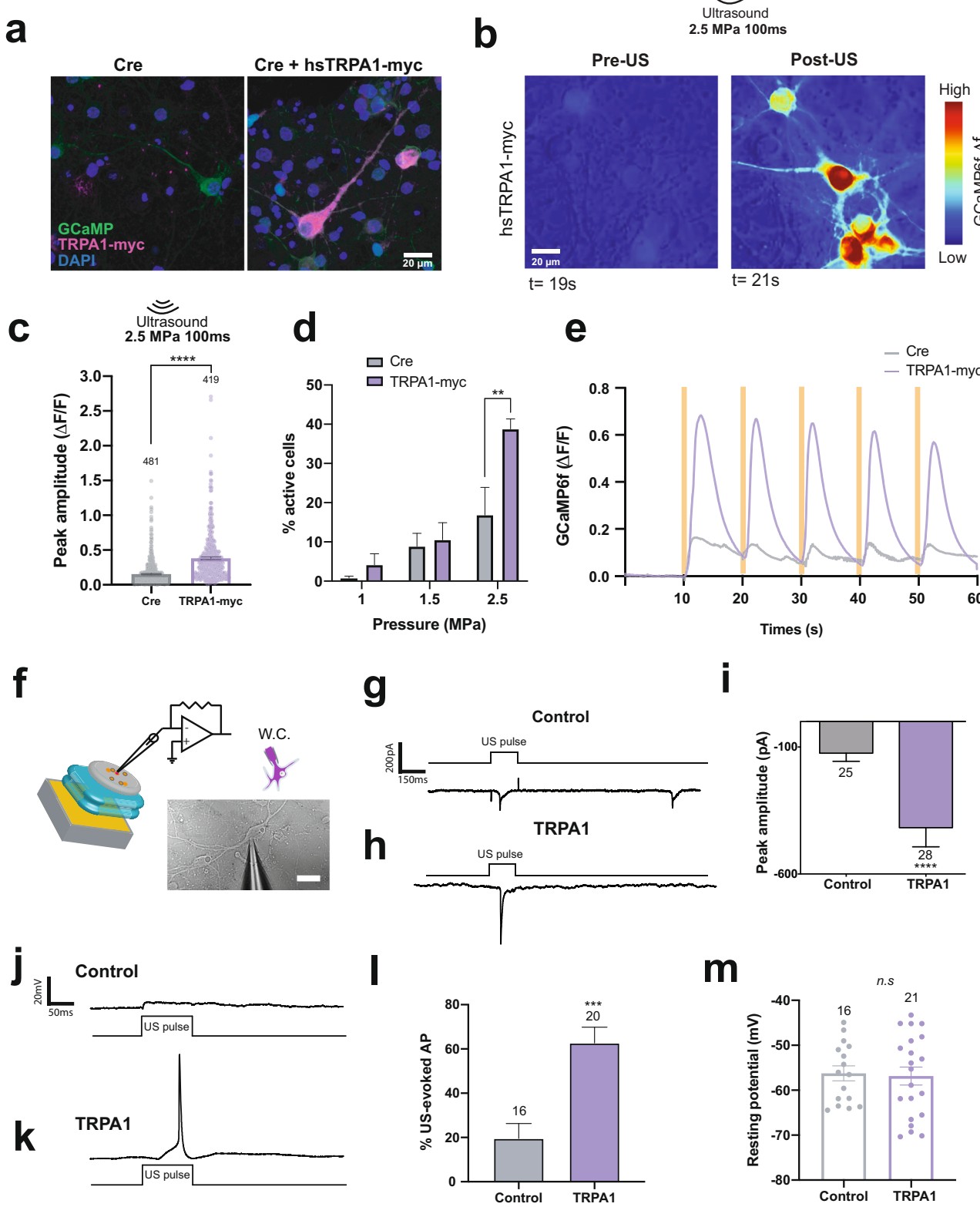

ultrasound-evoked action potentials, though not every ultrasound stimulus triggered an action potential, whereas control neurons exhibited infrequent action potentials in response to ultrasound (Fig. 4l). We focused our analyses specifically on response properties related to and evoked by ultrasound stimuli and not mechanosensitive ion channels expressed on neuronal

membranes. It is possible that the pressures delivered via pipette suction may activate these mechano-responsive ion channels, but we did not thoroughly test that hypothesis here. Nonetheless, we did not observe large differences in spontaneous activity between *hs*TRPA1 and Cre control neurons (Supplementary Fig. S9a) or resting membrane potential, suggesting that outside of ultrasound

**Fig. 4 *hs*TRPA1 potentiates calcium responses and evoked action potentials upon ultrasound stimulation in rodent primary neurons in vitro.**
**a** Representative images showing mouse primary neurons day in vitro (DIV) 12, expressing *hs*TRPA1 or controls. **b** GCaMP6f fluorescence in *hs*TRPA1 expressing neurons before and after ultrasound stimulus. Plots showing peak amplitude of GCaMP6f fluorescence upon **c**, 2.5 MPa ultrasound stimuli of 100 ms duration, and **d**, 100 ms stimuli at different pressures. **e** Average ratio of change in fluorescence to baseline fluorescence in neurons expressing *hs*TRPA1 or control Cre during repetitive 100 ms, 2.5 MPa ultrasound stimulation. Ultrasound is delivered every 10 s indicated by an orange bar. The number of GCaMP6f-expressing neurons analyzed is indicated above each bar. **c** ****$p < 0.0001$, by Mann-Whitney U test; **d** *$p < 0.05$, **$p < 0.01$, ****$p < 0.0001$ by two-way ANOVA with Geisser-Greenhouse correction. n = 3 coverslips/condition. **f** Schematic showing whole cell patch electrophysiology of rat primary neurons expressing *hs*TRPA1 used for both voltage-clamp and current-clamp recordings, whole-cell (W.C.) configuration. Representative gap-free voltage-clamp traces of **g**, control or **h**, neurons expressing *hs*TRPA1 upon 100 ms 7 MHz ultrasound stimuli in the 0.25 MPa range. **i** Plot showing peak amplitude response to ultrasound stimuli in neurons expressing *hs*TRPA1 or controls (Cre) ($N = 10$ and 12 neurons for TRPA1 and controls, respectively for panels i/l). Only one neuron was recorded per coverslip in order to avoid confounding effects from ultrasound stimulation trials on other neurons. 7 MHz 0.25 MPa ultrasound in DIV 11-14 rat primary neurons under current-clamp mode elicits subthreshold voltage changes in controls (**j**) and action potentials (**k**) in *hs*TRPA1 expressing cells. **l** Percent of trials in which an action potential was elicited by ultrasound in controls and *hs*TRPA1-expressing neurons. **m** Resting membrane potential is not altered in primary neurons upon expression of *hs*TRPA1. Data are mean ± SEM (**c**, **d**, **i**, **l**, **m**); n = 3 (**c–e**) biologically independent replicates and n = 10 and 12 neurons for TRPA1 and controls, respectively. Only one neuron was recorded per coverslip in order to avoid confounding effects from ultrasound stimulation trials on other neurons (**i**, **l**). n.s. $p > 0.05$, ***$p < 0.001$, ****$p < 0.0001$ by unpaired, two-tailed Mann-Whitney U test. Source data are provided with this paper.

stimulation, mechanical effects themselves were at a minimum. However, it remains to be determined how *hs*TRPA1 expressing dissociated neurons may respond to other modes and intensities of mechanical stimulation, Moreover, neither the latency, peak voltage, or time to peak of action potentials in response to ultrasound were altered by expression of *hs*TRPA1 (Supplementary Fig. S9g–i), and the membrane resting potential was similar between the two groups (Fig. 4m). Further, we found that we obtained comparable expression of exogenous *hs*TRPA1 channels in both mouse and rat primary neurons used for calcium imaging and electrophysiology studies respectively. (Supplementary Fig. S9j, k). These data demonstrate that ultrasound triggers increased current amplitudes, and thereby increases action potential probability in *hs*TRPA1-expressing neurons, even at ultrasound pressures at or below those shown to elicit responses by calcium imaging.

**hsTRPA1 confers ultrasound sensitivity in vivo.** We next determined whether *hs*TRPA1 can be used as a sonogenetic tool for temporally selective activation of neurons in vivo. To this end, we used Cre-dependent AAV to restrict the expression of *hs*TRPA1 to layer V motor cortical neurons in Npr3:Cre transgenic mice[46] (Fig. 5a). We first used in situ hybridization to confirm that adult cortical neurons do not express endogenous mouse TRPA1 (Supplementary Fig. SFig. 6). Consistently, data from the Allen Brain Atlas, Biogps and Brain-seq projects confirm that TRPA1 expression is undetectable in the brain[47–49]. Using coordinates based on a previous study[50], we co-injected AAV9 encoding myc-tagged *hs*TRPA1 with AAV9 encoding GFP to visualize the transfected neurons, into the left motor cortex. This approach robustly transduced layer V cortical neurons throughout the forelimb and hindlimb motor cortices (Fig. 5b and Supplementary Fig. S10a) and their projections in the right cervical and lumbar spinal cord (Supplementary Fig. S10b)[51]. Using the 7 MHz lithium niobate transducer coupled to the exposed skull through ultrasound gel, we verified our ability to deliver ultrasound to cortical regions. This in vivo approach delivered peak negative pressures ranging from 0.35–1.05 MPa with minimal temperature changes, despite the mouse skull attenuating 75% of the peak negative pressure from the transducer and the resulting stimuli generating different pressures in the cortex and other brain regions (Supplementary Fig. S11). These data suggest that we are also able to non-invasively target deeper brain regions with our ultrasound delivery system non-invasively.

After 2–4 weeks following intracranial injection, we monitored ultrasound-evoked electromyography (EMG) responses in the bilateral biceps brachii and biceps femorii muscles. Ultrasound evoked few EMG responses and no visible movements in any of the limbs of GFP-control mice (Fig. 5d–f). In contrast, animals injected with AAV9-*hs*TRPA1 in the left motor cortex showed dose-dependent EMG responses and visible movement in their right fore- and/or hindlimbs (Fig. 5d, e and Video S6). EMG responses in the left limbs occurred infrequently, suggesting circuit-specific sensitivity to ultrasound (Fig. 5f). Consistently, we also observed some of the transduced cortical neuron processes innervating the left forelimb motor pools (Supplementary Fig. S10d). Moreover, while most EMG responses occurred within one second of ultrasound stimulation, the latency and duration of these responses increased with stimulus duration (Fig. 5g, h). To confirm functional activation of cortical neurons, we then tested whether ultrasound stimulation increased *c-fos* in motor cortical neurons expressing *hs*TRPA1. While ultrasound stimulation had no effect on the number of *c-fos* positive cells in animals expressing GFP, it significantly increased the number of *c-fos* positive cells in cortical motor neurons of *hs*TRPA1-expressing mice (Fig. 6a–c and Supplementary Fig. S12b). This upregulation was specific to the cortex and we did not detect increased *c-fos* in the auditory cortex in these animals (Supplementary Fig. S12c–f), suggesting that the ultrasound-mediated effect does not involve incidental activation of the auditory cortex as has been previously suggested[14,15].

We could reliably activate neurons both in vitro and in vivo using sonogenetics. To assess its safety in vivo, we studied two metrics of safety, the effect of cortical TRPA1 expression on a motor learning task and the effect of sustained ultrasound delivery on the integrity of the blood-brain barrier. We found that both *hs*TRPA1- and GFP-expressing animals had comparable ability to learn the rotarod task (Supplementary Fig. S12a). Similarly, we found that animals receiving one hour of intermittent ultrasound stimulation had no damage to their blood-brain barrier. In contrast to stab wound positive control animals in which both 10 kDa fluorescent dextran accumulated and mouse IgG showed elevated binding, no increase in fluorescence was observed in animals receiving ultrasound, indicating that neither large nor small proteins were able to leak through the blood-brain barrier during sonication (Supplementary Fig. S13). Taken together, these results show that ultrasound can be used to selectively modulate neurons infected with AAV9-*hs*TRPA1 through an intact mouse skull at a frequency and pressure that neither affects normal behavior nor causes blood-brain barrier impairment.

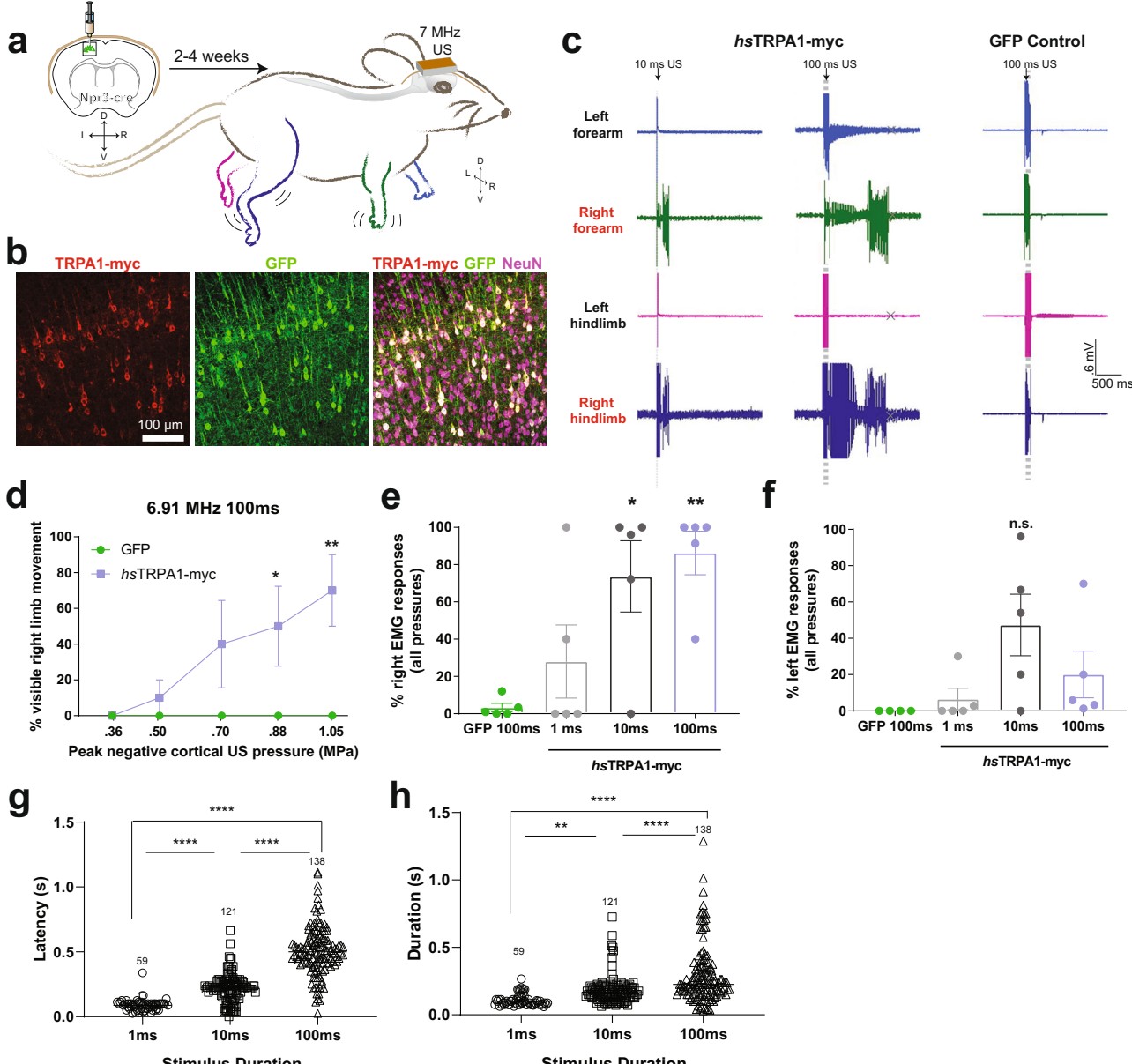

**Fig. 5 hsTRPA1 enables sonogenetic activation of mouse layer V motor cortex neurons in vivo. a** Schematic showing expression of *hs*TRPA1 or GFP controls in the left motor cortex of Npr3-Cre transgenic mice innervating the right fore and hindlimbs allowing these to be controlled with ultrasound stimuli. **b** Images showing expression of hsTRPA1 and GFP (co-injection marker) in layer 5 cortical neurons in vivo. Expression was evaluated in all experimental samples across multiple brai sections. **c** Representative EMG responses to 10 ms and 100 ms ultrasound stimuli from animals expressing *hs*TRPA1 and controls. **d** Visible right limb movements were scored in response to 100 ms ultrasound pulses of varying intensities. Plots showing percent of (**e**) right fore and hindlimb and (**f**) left fore and hindlimb EMG responses relative to number of stimulations pooled across all intensities. Plots showing (**g**) latency between the start of the ultrasound pulse and subsequent EMG response, and (h) duration from the start of the EMG response until the signal returned to baseline. **d–f** Data are mean + - SEM, $n = 5$ biologically independent animals/group, **g–h**, $n = 59$–138 responses collected from successful trials in $n = 5$ biological animals. *$p < 0.05$, **$p < 0.01$, ****$p < 0.0001$ compared to GFP control by two-way ANOVA followed by Tukey's multiple comparisons. Source data are provided with this paper.

## Discussion

We demonstrate that *hs*TRPA1 is a candidate sonogenetic protein that confers ultrasound sensitivity to mammalian HEK cells and rodent neurons in vitro and in vivo (Fig. 6d). Using an unbiased screen, we found that *hs*TRPA1-expressing HEK cells show ultrasound-evoked calcium influx and membrane currents. Moreover, we reveal critical components of *hs*TRPA1 ultrasound sensitivity, including the N-terminal tip region and interactions with the actin cytoskeleton and cholesterol. We also show that *hs*TRPA1 potentiates ultrasound-evoked calcium transients and

enables ultrasound-evoked action potentials in rodent primary neurons. This uses patch clamp to monitor ultrasound-induced action potentials at clinically relevant frequencies, lower than 25 MHz. In addition, we used *hs*TRPA1 to selectively activate neurons within an intact mouse skull, using pulses of ultrasound ranging from 1–100 ms. These ultrasound parameters are below the range associated with cavitation[52]. Accordingly, we observed no damage to the blood-brain barrier even with intermittent ultrasound delivered over 60 min. Moreover, overexpressing *hs*TRPA1 did not cause behavioral changes on rotarod assays,

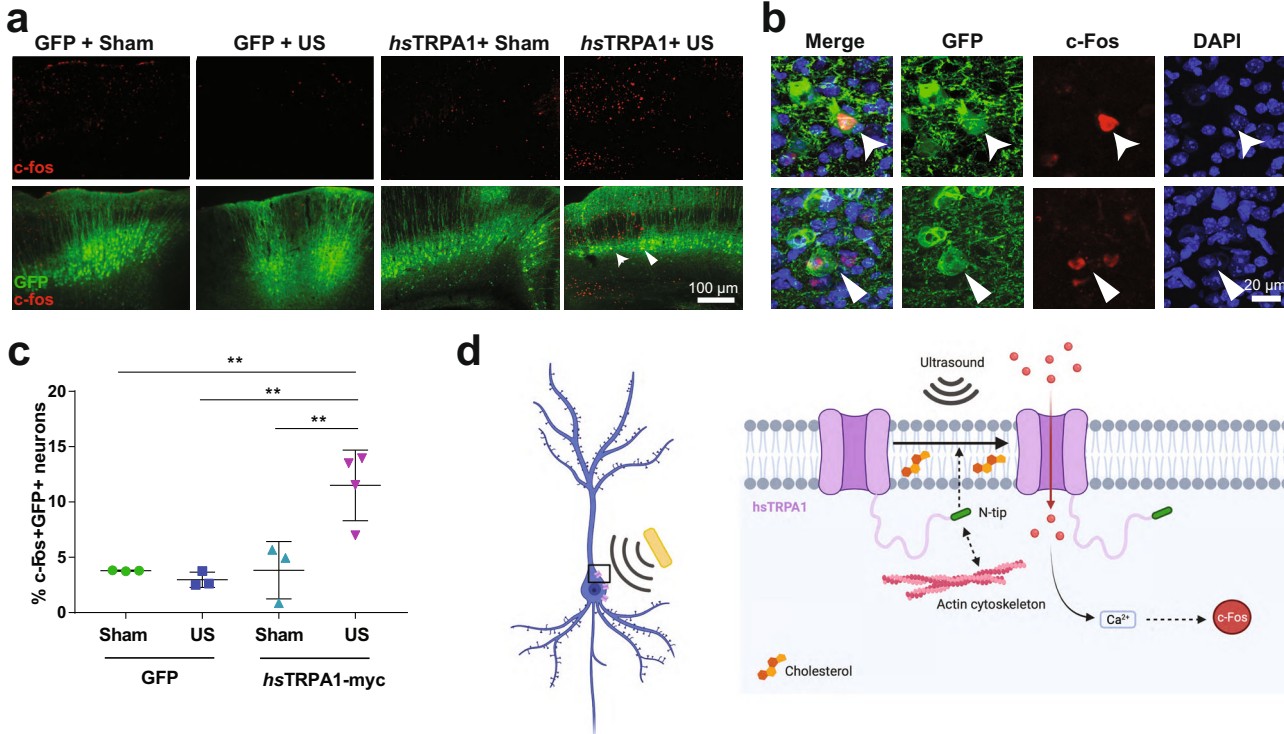

**Fig. 6 *hs*TRPA1 enables ultrasound-triggered *c-fos* activation in vivo. a** Representative images from sections taken at ~700 µM intervals throughout the GFP + region of the cortex with c-fos staining. Expression of c-fos and GFP were quantified across all sections with positive GFP immunolabeling in each animal. Zoomed images showing co-localization of **b**, *c-fos* and GFP and *c-fos* and DAPI within GFP positive neurons. **c** Quantification of *c-fos* positive, GFP + neurons in different experimental groups, after ultrasound stimulation. Data are mean + - SEM, n = 3–4 biologically independent animals/group **p < 0.01 compared to GFP control by two-way ANOVA followed by Tukey's multiple comparisons. Source data are provided with this paper. **d** Proposed model of action of *hs*TRPA1 in neurons, illustrating requirement of the N-terminal tip, actin cytoskeleton, and cholesterol to induce an ultrasound-dependent increase in intracellular calcium resulting in neuronal activation and *c-fos* expression.

confirming the viability of this candidate protein for sonogenetics use in rodent experiments. Our results are in contrast to a previous study showing that mouse TRPA1 in astrocytes responds to ultrasound through a Best1 dependent pathway to release glutamate and depolarize neighboring neurons[13]. That model requires TRPA1 and Best1 expression in astrocytes, which is highly controversial[53–55]. Additionally, we and others confirm that TRPA1 is undetectable in the brain[47–49] and that ultrasound can non-invasively activate neurons expressing exogenous hsTRPA1 in vitro and in vivo, indicating the viability of our method.

Ultrasound has been shown to have neuromodulatory effects in mice[4,5], non-human primates[6,7], and even human subjects[9,10], though the underlying mechanisms remain poorly understood. We previously showed that overexpressing the mechanosensory receptor TRP-4 (a TRP-N homolog) in *C. elegans* neurons renders them sensitive to short pulses of ultrasound, identifying the first putative sonogenetic candidate[18]. Multiple groups have since identified additional ultrasound-sensitive candidates including MscL[19], Prestin[21], Piezo[20], TREK[22], MEC-4[17], TRPC1, TRPP2, and TRPM4[16] using in vitro assays. Of these, a mutated form of MscL has demonstrated activity in vivo[56], but there is nevertheless value and a need to identify novel sonogenetic candidates to extend the toolset and potentially develop channels that respond to different ranges of frequency and pressure. We identified *hs*TRPA1 and its mammalian homologs as top hits for high-frequency sonogenetic candidates in an unbiased screen from a curated library of candidate proteins, emphasizing the unique nature of this protein. In addition, we find that *hs*TRPA1 outperforms previously identified sonogenetic candidates across different ultrasound frequencies.

Previous studies have shown that ankyrin repeats form a superhelical coil that could act as springs for mechanosensitive gating in NOMPC/TRPN1[36]. We show that the TRPA1 N-terminal tip domain, particularly the first 25 amino acids, might be critical for ultrasound sensitivity and is highly similar in mammalian TRPA1 variants that showed sensitivity to ultrasound, but varies across non-mammalian chordate TRPA1 homologs that did not. Furthermore, a chimera composed of the *am*TRPA1 N-terminal tip on *hs*TRPA1 also lacks responses to ultrasound, providing stronger evidence that this region is important for tuning ultrasound sensitivity in mammalian TRPA1 variants. These results are consistent with a critical role for N-terminal tip in modifying the temperature sensitivity of TRPA1[57]. However, additional reports have shown that this N-terminal domain is not essential for sensing temperature changes[58,59], but instead affects channel function by modulating the actin cytoskeleton[58,60]. We further show that an intact actin cytoskeleton is required for *hs*TRPA1 ultrasound responses. Consistently, previous studies have shown that the actin cytoskeleton can either directly interact with mechanosensitive channels[61] or interact with the plasma membrane to modify mechanosensation[62]. We therefore speculate that the *hs*TRPA1 N-terminal tip region may interact with the actin cytoskeleton to transduce ultrasound-induced membrane perturbations into changes in intracellular calcium. Consistent with our results, a recent study found that *hs*TRPA1 is intrinsically mechanosensitive, and suggests a role for the N-terminal domain in modifying these responses[60]. Our analysis of TRPA1 sequences across homologs further suggests that a CRAC domain that is thought to mediate interactions with cholesterol[42] is heavily

modified or missing from the second transmembrane domain of ultrasound-insensitive variants. Indeed, we found that interaction with the lipid bilayer is critical for ultrasound sensitivity of hsTRPA1 as treating hsTRPA1-expressing cells with methyl-β-cyclodextrin, which removes cholesterol, attenuated their responses to ultrasound but not to a chemical agonist. Furthermore, mutation of the central tyrosine that is critical for cholesterol interaction of the CRAC domain likewise impaired ultrasound sensitivity, but not responses to AITC. These data are consistent with previous studies showing that TRPA1 activation requires membrane lipid interactions[63].

Previous studies have shown that naïve neurons can respond to ultrasound both in vitro[16] and in vivo[4]. Similarly, we observe that the ultrasound parameters we use can also trigger increased currents and intracellular calcium in naïve neurons in vitro. However, these responses are significantly smaller than those observed in hsTRPA1-expressing neurons, and ultrasound-evoked action potentials were rarely detected at the frequency and pressures tested. Additionally, the infrequent responses in control neurons in vitro could be a result of membrane deflection as we recently showed[64], or a result of interactions with the substrate, or with the patch pipette in electrophysiology[65]. Control neurons have been previously shown to generate action potentials upon ultrasound stimuli at lower fundamental frequencies, however, these responses required repeated pulses[4]. Further experiments using more physiologically relevant systems, such as brain slices and 3-dimensional neuronal cultures will be needed to explore the extent of the intrinsic neuronal sensitivity to ultrasound at 7 MHz. Moreover, we find that ultrasound responses in control neurons are unlikely to be endogenous TRPA1-mediated, as these are not reduced upon treatment with TRPA1 antagonists, and because neurons cultured from TRPA1-/- mice also responded to ultrasound. Accordingly, we did not detect endogenous TRPA1 in E18 brain tissue. Together, these results suggest that intrinsic neuronal responses to our ultrasound parameters are unlikely to involve TRPA1 in neurons. Instead, a recent study found that knocking down TRPP1, TRPP2, Piezo, TRPC1, and TRPM4 each partially reduce ultrasound-evoked neuronal responses[16]. We also show that blocking voltage-gated sodium channels eliminated neuronal ultrasound-evoked calcium responses. Therefore, intrinsic ultrasound neuromodulation may involve a number of ion channels whose activity is further amplified by voltage-gated sodium channels. We show that hsTRPA1-expressing neurons maintain partial ultrasound sensitivity in the presence of a sodium-channel blocker, confirming that hsTRPA1-mediated ultrasound sensitivity is at least partially independent from the mechanism contributing to ultrasound activation in control neurons.

Finally, we demonstrate that hsTRPA1 can be used to selectively activate a specific cell population in vivo with ultrasound pulses (1–100 ms) from a 7 MHz transducer. Our data hint that ultrasound might not act as a simple stretch force on the membrane, and that channels that likewise sense other perturbations, including lipid bilayer changes, could be good candidates for sonogenetics. Moreover, our identification of specific interactions (namely, actin, and membrane cholesterol) and the N-terminal tip domain in hsTRPA1 will allow us to rapidly engineer this channel for enhanced ultrasound sensitivity and ion permeability. Broadly speaking, we suggest that hsTRPA1 and its variants could be used to noninvasively control neurons and other cell types across species.

## Methods

**Ethical Statement**. Our study complies with all relevant ethical regulations established by the Institutional Biosafety committee of the Salk Institute for Biological Studies.

**Animal husbandry**. Studies were performed using a total of 50 adult mice, including both males and females. Animals were group housed in an American Association for the Accreditation of Laboratory Animal Care approved vivarium on a 12-h light/dark cycle between 68 and 72⁰F and humidity of 30–70%, and all protocols were approved by the Institutional Animal Care and Use Committee of the Salk Institute for Biological Studies. Food and water were provided ad libitum, and nesting material was provided as enrichment. Colonies of C57BL/6J (JAX# 000664), Npr3-cre[46] (JAX# 031333), TRPA1 knockout[66] (JAX #006401) mice were maintained for experiments.

A total of 50 mice were used in our studies.

For mouse neuronal cultures: WT E18 embryos were collected from timed-pregnant C57BL/6J mice (JAX# 000664) (n = 3 pregnant mice total). TRPA1-/- E18 embryos were collected from a TRPA1-/- (JAX #006401) female bred with a TRPA1-/- male mouse (n = 2 pregnant mice total).

Hydrophone measurements were taken in 1 18 week-old male Npr3-cre mouse, 1 12 week-old female Npr3-cre mouse and 1 12-week-old Bl/6 male mouse (n = 3 mice total). Brain and DRG tissue for Basescope analysis was taken from an 11 week-old female C57BL/6J mouse (JAX# 000664), a 20 week-old female TRPA1 -/- mouse (JAX #006401), and an E18 embryo taken from a timed-pregnant C57BL/6J mouse (JAX# 000664). Homozygous Npr3-cre mice (JAX# 031333) received cortical viral injections between 12 and 20 weeks of age. EMG data were collected between 2 and 4 weeks after injection. Data presented are from n = 2 males and n = 3 females for GFP and n = 1 males, and n = 4 females for hsTRPA1-injected mice. (n = 10 mice total). For rotarod and c-fos experiments, homozygous Npr3-cre mice (JAX# 031333) received cortical viral injections between 10 and 20 weeks of age. Rotarod data were collected 2 weeks after injection. c-fos data were collected 6 weeks after injection. Delays in c-fos data collection were in part due to the Covid-19 shutdowns. N = 5 males and n = 2 females were included in GFP and hsTRPA1-injected groups. (n = 14 mice total). For blood-brain barrier experiments age-matched Balb/c mice (JAX # BALB/cJ/000651) were used in experiments at 9 weeks of age. All groups included n = 2 male and n = 2 female mice, excepting the stab wound condition, which included n = 2 male and n = 3 female mice. (n = 17 mice total).

**BaseScope**. Adult TRPA1 knockout[66] (JAX #006401) and wild-type C57BL/6J (JAX #000664) mice were perfused with 0.9% saline. A WT C57BL/6J E18 mouse embryo was also collected from a cohort of embryos slated for dissociation for use in in vitro experiments. Brains and lumbar DRG were extracted and immediately frozen in OCT. Fresh frozen sections (10um) were direct mounted and slides were stored at −80 °C overnight. Tissues were labeled with a custom BaseScope probe (BA-Mm-Trpa1-3zz-st, ACD-Bio Probe Design #: NPR-0003309) targeting 2602–2738 of mouse TRPA1 (NM_177781.5. GTGATTTTT AAAACATTGC TGAGATCGAC CGGAGTGTTT ATCTTCCTCC TACTGGCTTT TGGCCTCA GC TTTTATGTTC TCCTGAATTT CCAAGATGCC TTCAGCACCC CATTG CTTTC CTTAATCCAG ACATTCAG). This region was chosen because it is deleted in the TRPA1 knockout mouse. Tissues were also probed with positive (Ppib, ACD #701071) and negative control (DapB, ACD#701011) probes, which gave the expected results in all experiments. DRG tissue was used as positive control for the TRPA1 probe, as small-diameter DRG neurons are known to express TRPA1. DRGs and cortices from TRPA1 knockout mice were used as a negative control for the TRPA1 probe.

**Blood-brain barrier experiments**. Mice received retro-orbital injections of 10 kDa fluorescein isothiocyanate-dextran (Sigma FD10S) at 150 mg/mL in saline. Positive control mice immediately received a cortical stab wound with a 27 g needle through a small hole drilled at AP 0, ML −1, DV −0.5. Ultrasound-treated mice had their scalp opened and the ultrasound transducer was coupled over the left cortex with ultrasound gel. The transducer delivered 100 ms stimuli at 0.88 MPa every 10 s for 1 h. Sham-treated mice underwent the same procedure but the transducer was not turned on. A cohort of mice that did not receive dextran injection or ultrasound was also collected as a negative control, and all values were normalized to this cohort. (n = 4–5 per group, evenly split between male and female). Mice were perfused 70 min after cortical injection or the start of the ultrasound treatment. One ultrasound-treated mouse had to be omitted from the data set due to inadequate perfusion. Brain sections were sectioned at 35 μM and processed as floating sections. After blocking, sections were incubated with 647 donkey anti-mouse and DAPI. Images were collected on an Olympus Virtual Slide Scanner (VS-120), using the same settings across all groups. Fluorescein isothiocyanate-dextran 488 and 647-mouse IgG were quantified as mean intensity in the left and right cortex from each sample using Fiji[67]. All values were normalized to fluorescent values obtained from samples that received neither dextran nor ultrasound.

**Calcium imaging analysis**. All image analysis was performed using custom scripts written as ImageJ Macros. Cells in the dTom channel were segmented and cell fluorescence over time in the GCaMP channel was measured and stored in.csv files. Briefly, the script uses a Gaussian filter on the dTomato channel and background subtraction, followed by auto thresholding and watershed segmentation. The plugin 'Analyze particles' was then use to extract counts. Calcium data were analyzed using custom Python scripts. Calcium signal was normalized as ΔF/F

using a 6 s baseline for each ROI and a peak detection algorithm with a fixed threshold of 0.25 was used to identify responsive cells after ultrasound stimulation, similar to the approached used by[13]. For the screen, the number of cells showing a response to ultrasound was calculated as the total percent of responsive cells after 3 consecutive 90-second recordings on the same coverslip. The percent of transfected cells was calculated as the number of dTom positive cells/total number of cells per field of view imaged. To compare ultrasound response between clones, we used a generalized mixed model, fitting "response" as a Bernoulli response, "clone" as a fixed factor and "cell" as a random effect. Pairwise comparisons were later performed using odds ratios and the Tukey method, correcting for multiple comparisons.

Peak amplitude was calculated for each trace as the maximum GCaMP6f ΔF/F value during 60 s after ultrasound stimulation or pharmacological treatment for HEK cells, and 5 s for mouse primary neurons. For the AITC response curve in neurons, mean GCaMP6f ΔF/F up to 1.5 min after adding AITC to the media was used instead of peak amplitude response. For latency and duration analysis in primary neurons, the latency of calcium responses was measured as the time to reach 63% of the peak amplitude after stimulation, while the width was calculated as the distance between 63% rise and 63% decay.

**Candidate channel screen.** We generated a library of candidate channels that was initially based on a literature survey of naturally occurring ion channels and other membrane proteins that have been suggested to display mechanosensitive or ultrasound sensitive properties[18–21,68,69]. From this initial list, related channels and variants from different species were selected, resulting in a final set of 191 proteins (Supplementary Table S1). Each channel was codon optimized for expression in human cells and cloned into a custom bicistronic pcDNA3.1(+) vector using a porcine teschovirua-1 2 A self-cleaving peptide (p2A) sequence, expressing the channel and the fluorescent protein dTomato under a human cytomegalovirus (CMV) promoter. All plasmids were generated by Genscript Biotech (New Jersey, United States).

**Consensus sequence and percent identity.** Consensus sequences for the 10 tested chordate, mammalian, and non-mammalian alignments, each having *hs*TRPA1 as reference, were generated in Geneious Prime. The threshold for consensus was set to 65%, as this ensured contribution from both mammalian and non-mammalian sequences for chordate, from rodents, ungulates, and bats for mammalian, and from reptiles and fishes for non-mammalian alignments. Alignment and consensus sequences were annotated in Geneious Prime to highlight either agreement or disagreement of a given amino acid relative to human TRPA1. Percent identity of consensus sequence to human was calculated to quantify the degree of sequence conservation or divergence in the chordate, mammalian, non-mammalian species.

**Constructs of ankyrin TRPA1 mutants.** To generate mutant constructs, a PCR-based approach was used. Bicistronic constructs co-expressing deletion mutants and dTom were synthesized by Genscript Biotech (New Jersey, United States). For ΔANK1, amino acids (aa) 67–95 (Ref seq. UniProtKB - O75762) were deleted, corresponding to nucleotide deletions 2641–2727 (Ref seq. XM_017013946.1). For the rest of the constructs the aa and nucleotide deletions were as follows: ΔN-tip: aa deletions 1–61, nucleotide deletions 1–182; ΔN-tip (1–25), aa deletions 1–25, nucleotide deletions 1–75; CRAC mutant, swapping Tyr (Y)785 to Ser (S), nucleotides 2353–2355 (TAC) to TCG; *alligator* N-tip, swapping aa 1–66 from hsTRPA1 to first 66 residues from amTRPA1; *zebrafish* N-tip, swapping aa 1–59 from hsTRPA1 to first 59 residues from drTRPA1.

**CRAC-CARC motif annotation.** CRAC ([LV]X(1,5)YX(1,5)[RK]) and CARC ([RK]X(1,5)[YF]X(1,5)[LV]) motifs, as defined in[70] were annotated per TRPA1 sequence using the Geneious Prime EMBOSS 6.5.7 fuzzpro tool[71].

**EMG experiments.** EMG experiments were conducted between 2–4 weeks after viral injection. EMG data were collected under ketamine (100 mg/kg) and xylazine (10 mg/kg) anesthesia from the right and left biceps brachii and right and left biceps femoris through fine wire electrodes (A-M Systems 790700) connected to a PowerLab and BioAmp (AD Instruments). Data were collected at 40k/sec, band-pass filtered from 300 Hz to 1 kHz. Correct electrode placement was confirmed by a positive EMG signal in response to the pinch. The skin over the skull was opened, and the 7 MHz lithium niobate ultrasound transducer was coupled to the skull using ultrasound gel (Parker Aquasonic 100). Ultrasound stimuli (1, 10, 100 ms durations) were administered at no less than 10 s intervals at intensities ranging from 0.35–1.05 MPa, intracortical pressure. Visual movement of the right fore or hindlimb in response to stimulation was noted and EMG responses were analyzed for latency and duration. Due to the relatively large stimulus artefact from the ultrasound pulse, responses occurring during the ultrasound stimulus could not be reliably quantified. Therefore, only responses occurring after cessation of the stimulus were considered in our analyses. The experimenter was blinded as to the group during both collection and analysis of the data.

**HEK cell culture and transfection.** HEK cells expressing human αvß3 integrin[72] were cultured in DMEM supplemented with 10% FBS and 20 mM glutamine in a 5% $CO_2$ incubator. A stable calcium reporter line was generated with a GCaMP6f lentivirus (Cellomics Technology PLV-10181-50) followed by FACS sorting. For screening experiments and characterization of each candidate channel, GCaMP6f-expressing HEK cells were seeded on 12-well cell culture plates with 18 mm glass coverslips coated with PDL (10 μg/μl; Sigma-Aldrich P6407) for 1-2 h. Coverslips were washed with Milli-Q water and cells seeded at a density of 250000 cells/well. Cells were transfected with Lipofectamine LTX Reagent (ThermoFisher 15338100) according to the manufacturer's protocol and 24 h after plating, using 500 ng DNA of the clone of interest for each well. Cells were kept at 37 °C for an additional 24 h before imaging on our ultrasound stimulation setup.

**Imaging rig for ultrasound stimulation.** We upgraded an existing upright epi-fluorescent Zeiss microscope to perform a monolayer two-dimensional screen. For this application, we used our custom-made 12x12 mm lithium niobate transducer placed in a heated stage fixture underneath the cell chamber. Stimulus frequency and duration were controlled by a waveform generator (Keysight 33600 A Series), and pressure was controlled through a 300-W amplifier (VTC2057574, Vox Technologies, Richardson, TX). Simultaneous calcium imaging was performed using a 40x water dipping objective at 16.6 frames per second with an Orca Flash 4.0 camera and a GFP filter.

**Immunocytochemistry.** Cells were fixed with 4% paraformaldehyde (PFA) at room temperature for 15 min and subsequently permeabilized by 0.25% Triton X-100 PBS with 5% horse serum. After incubation in blocking solution for 1 h at room temperature, cells were incubated overnight at 4 °C with different primary antibodies: for HEK cells a mouse monoclonal αTRPA1 antibody (1:1000; Santa Cruz Biotech #376495) was used, while a rabbit polyclonal anti-myc antibody (1:1000; Cell Signaling Tech #2272 S) was used to detect the tagged channel in primary neuron cultures. Secondary antibody staining was performed at room temperature for 2 h, followed by DAPI for 30 min. For myc, TSA amplification was performed to increase the signal[73]. Colocalization to the cell membrane was determined via co-transfection and co-immunolabeling with EGFP-CAAX[74], which was a gift from Lei Lu (Addgene plasmid #86056; http://n2t.net/addgene:86056; RRID:Addgene_86056). For cytoskeleton immunolabeling experiments, fixed cells were incubated with alpha-tubulin antibody (Sigma, #CBL270-I, 1:1000) or phalloidin-488 (ThermoFisher, #A12379, 1:500).

**Immunohistochemistry and *c-fos* quantification.** At the conclusion of the study, mice were perfused with 0.9% saline followed by 4% paraformaldehyde (PFA) through a peristaltic pump. Brain tissue was immediately collected and incubated in 4% PFA overnight before being changed to 30% sucrose. The tissue was then sectioned at 35 μM into tissue collection solution (glycerine, ethylene glycol, $NaH_2PO_4$, $Na_2HPO_4$) and stored at 4 °C. For brain immunohistochemistry, brain sections from ~every 350 μM were immunolabeled for myc (1:500; Cell Signaling 2272 S), c-fos (1:500; Encor RCPA-cfos), NeuN (1:1500; Synaptic Systems 226004), GFP (1:1000; AVES GFP-1010) and DAPI (1:1000). Tyramide amplification was used to enhance the myc and c-fos signals. Briefly, tissue was incubated for 30 min in $H_2O_2$, blocked for 1 hr in PBST plus 5% horse serum, and then incubated overnight with primary antibodies. The next day, tissue was incubated for 3 hrs at room temp with biotinylated donkey anti-rabbit (1:500, Jackson Immunoresearch 711-065-152), then washed, incubated with ABC (Vector Labs PK-4000) for 30 min, washed, incubated with tyramide[73], washed and incubated with streptavidin-conjugated antibody along with secondaries (ThermoFisher Scientific and Jackson Immunoresearch) appropriate to other antigens of interest for 3 h at room temp or overnight at 4 °C. Tissue was then mounted onto glass slides and coverslipped with Prolong Gold Antifade mounting medium (ThermoFisher Scientific). Imaging for quantification of *c-fos* and myc expression were conducted at 10x on a Zeiss Axio Imager.M2 connected to an OrcaFlash 4.0 C11440 camera. High-quality images depicting myc and fos colocalization with GFP were taken on a Zeiss Airyscan 880 microscope. Imaging of whole-brain sections was conducted at 10x on an Olympus VS-120 Virtual Slide Scanning Microscope.

Quantification of c-fos and GFP positive neurons was conducted in Fiji[67] using manual cell counting. c-fos puncta were excluded if they did not colocalize with DAPI. Myc+ GFP + neurons were also quantified using manual cell counting in Fiji. Only GFP + cell bodies that were completely filled with myc immunolabeling were considered to be myc + . The experimenter was blinded as to the experimental condition during quantification.

**In vitro electrophysiology.** A stable line of HEK cells expressing Nav1.3 and Kir2.1 (Ex-HEK[30], ATCC CRL-3269) were cultured on 18 mm round coverslips, at a seeding density of ~300k cells/well in a tissue-culture treated 12-well plate. Cells were transiently transfected with a custom plasmid (Genscript) expressing hsTRPA1 and dTom fluorescent reporter as for screening experiments, 18-24 h postseeding. Cells underwent a media change, were allowed to recover, and then were used for recordings 18-24 h after transfection. Coverslips were transferred to a custom machined acrylic stage containing a bath of external solution; NaCl (140 mM), KCl (4 mM), $CaCl_2$ (2 mM), Glucose (5 mM), and HEPES (10 mM)

with an osmolarity of ~295 mOsm. Patch pipettes were pulled on a Sutter puller model P-97 programmed to give 4–6 MΩ tips from filamented borosilicate glass (o.d. 1.5 mm, i.d. 0.86 mm). These pipettes were filled with an internal solution [110 mM KF, 10 mM NaCl, 10 mM KCl, 10 mM EGTA, 10 mM HEPES adjusted to a pH 7.2 using KOH with an osmolarity of ~285 mOsm (Nan]i[on, #08 3007)]. An Olympus 40x water dipping lens with 0.8 NA was used in combination with a (QImaging OptiMOS) cMOS camera used to visualize cells with Köhler or fluorescent illumination. dTom signal was used to confirm *hs*TRPA1 expression in HEK cells. Electrical signals were acquired using Axon Instruments Multiclamp 700B amplifier and digitized with Digidata using pClamp acquisition and control software. Gap-free recordings were conducted (typically holding the membrane potential at -70 mV) while delivering 100 ms pulses of ultrasound. The ultrasound delivery rig used for patch-clamp experiments was similar to the one used for imaging experiments. Briefly, waveforms were programmed using an arbitrary function generator (Keysight Technologies) connected via BNC to an amplifier (VTC2057574, Vox Technologies). Military communications grade BNC cables (Federal Custom Cable) were used to ensure impedance matching in our systems and reduce electrical interference. The amplifier was connected to our custom-made lithium niobate transducer mounted on a dove-tail sliding arm, and coupled to the bottom of the recording chamber with ultrasound gel. We deliberately chose 18 mm glass coverslips to ensure optimal coverage by our custom manufactured transducers. This permitted a visual search for morphologically healthy neurons that were located directly over the transducer. In order to maintain optimal and consistent ultrasound delivery to cultured cells, we built a custom acrylic perfusion chamber to fit a rigid sliding dovetail bridge manifold that held the transducer directly below the chamber and could be reliably positioned in the optical axis. The center of the transducer was left uncoated with gold in order to permit bright-field light to reach the sample, allowing us to align optics and obtain even illumination for DIC imaging. Recordings were carried out in response to peak negative pressures ranging from 0.2–0.25 MPa, as access resistance could not be maintained when high pressures were delivered. Cell attached Ex-HEK-GCaMP cells-maintained membrane resistances between 0.5 and 3 GΩ. Each cell had a recording duration of 5 min, unless membrane resistance stability was lost. Fig. 2e, I/Imax was computed from the steady-state current response after a depolarizing voltage step ($-120$mV to $+120$ mV). Imax represents the largest current response obtained from a voltage step for each recorded cell.

Patch-clamp experiments conducted on primary dissociated cortical neurons followed a modified protocol. Neurons were allowed to mature for 11–14 days in vitro prior to recording. Compared to HEK cells, neuron somatic morphology was better suited for whole-cell recording configuration. Both voltage-clamp (VC) and current clamp (CC) recordings were conducted. Upon successful whole-cell access, baseline gap-free recordings in CC or VC trials were obtained. Ultrasound stimulation parameters followed the same protocol as for the HEK cell recordings. All gap-free recordings lasted 5 min and had between 1 and 10 ultrasound stimulus trials. More than 1 trial was recorded if membrane resistance was maintained throughout the duration of ultrasound stimuli. For primary cortical neuron experiments, access resistance during successful whole-cell recordings was maintained between 10 and 25 MΩ.

**In vitro pharmacology**. For inhibition of TRPA1, we incubated cells with the antagonist HC-030031[13] (40μM in DMSO; Cayman Chemicals #11923) for 45 min before stimulation. For activation of TRPA1, we used NMM[13] (N-Methylmaleimide; 100μM in DMSO Sigma-Aldrich #389412) or AITC[28] (30μM in DMSO, allyl isothiocyanate; Sigma-Aldrich # 377430). For activation of Piezo1, we used yoda-1[75] (10 μM in DMSO; Tocris #5586). For activation of TRPV1 we used capsaicin[76] (3 μM in DMSO; Sigma-Aldrich #M2028). The final concentration of DMSO in the external solution was 0.1% or lower for all groups, which was also used as vehicle control. For cytoskeleton experiments, nocodazole (5μM; Tocris, #1228), jasplakinolide (200 μM; ThermoFisher # J7473), paclitaxel (600 nM; Sigma-Aldrich # T7191), cytochalasin D (5μM; Cayman Chemicals, #11330) or latrunculin A (1 μM; Cayman Chemicals, # 10630) in 0.1% DMSO were added to the culture media 45 min prior to imaging[37]. For pharmacology in primary neurons, we used TRPV1 antagonist, A784168[77] (20μM; Tocris, #4319, 45 min incubation) in 0.1% DMSO, BAPTA[78] (30μM; Invitrogen, #B1204, 45 min incubation) directly dissolved in culture media and TTX[79] (18μM; tetrodotoxin citrate; Tocris #1069, 5 min incubation, where we also inhibit TTX-R channels).

**Mouse primary embryonic neuron culture**. For WT primary neuron culture, timed pregnant C57BL/6J female mice were ordered for E18 cortical dissociation (Charles River: 027). For TRPA1 knockout neuron culture, female TRPA1-/- (JAX #006401) dams were injected with luteinizing hormone-releasing hormone (Sigma-Aldrich, L8008) 5 days before being paired with -/- males overnight. Pregnant dams were sacrificed and the E18 embryos were collected for cortical dissociation.

Mouse primary neuronal cultures were prepared from cortices isolated from embryonic day 18 (E18) mice, following the protocol described in[80]. Neurons were plated in 12-well culture plates with 18 mm PDL-coated coverslips (Neuvitro Corporation GG-18-PDL) at a concentration of 600–900k cells/well. Neurons were then incubated at 37 °C, 5% CO$_2$, with half media changes every 2-3 days with Neurobasal (ThermoFisher #21103049 supplemented with Primocin (InvivoGen #ant-pm-1), B-27 (ThermoFisher #17504044) and GlutaMAX (ThermoFisher

#35050061). For calcium imaging experiments, cells were infected with AAV9-hSyn-GCaMP6f (Addgene #100837-AAV9) at day in vitro 3 (DIV3) and half media change was performed the next day.

Neurons infected with GCaMP6f as stated above were infected with AAV9-hSyn-Cre (Addgene #105553-AAV9) and AAV9-hSyn-TRPA1-myc-DIO (Salk GT3 core) at DIV4 and half media changed was performed the next day. Cultures were incubated at 37 °C, 5% CO$_2$ until DIV10-12 and then imaged using the same equipment as for HEK cell experiments.

**Phylogenetic analysis**. A multiple sequence alignment of all ten TRPA1 sequences was generated using Geneious Prime MAFFT (version 7.450)[81], with a BLOSOM 62 scoring matrix, gap open penalty of 1.53, and offset value of 0.123. A phylogenetic gene tree based on the MAFFT alignment was generating using Geneious Prime RAxML (version 8.2.11)[82], with a GAMMA BLOSOM 62 protein model, bootstrapping using rapid hill-climbing with seed 1, starting with a complete random tree, and using the maximum likelihood search convergence criterion. The maximum likelihood tree was assessed and annotated in FigTree (version 1.4.4).

**qPCR in mouse and rat primary neurons**. RNA was extracted from primary neuronal cultures using Qiagen RNEasy mini kit (#74104), while cDNA synthesized using High-capacity RNA-to-cDNA kit (ThermoFisher #4387406) and processed for qPCR using an Applied Biosystems QuantStudio PCR system. Primers used for thermocycling were: Forward 5'-GTG GAA CTT CAT ACC AGC TTA GA -3' and Reverse 5'-AGA TCT GGG TTT GTT GGG ATA C -3'. Briefly, media was removed and cells immediately lysed by trituration with lysis buffer. Extracted RNA concentration was quantified using a NanoDrop (ThermoFisher). PCR reactions used Fast Advanced Master Mix (ThermoFisher #4444557) and run up to 40 cycles. Relative expression was normalized to the housekeeping gene GAPDH.

**Rat primary neuron culture**. Rat primary neuronal cultures were prepared from rat pup tissue at embryonic day (E) 18 containing combined cortex, hippocampus and ventricular zone. The tissue was obtained from BrainBits (Catalogue #: SDEHCV) in Hibernate-E media and used the same day for dissociation following their protocol. Briefly, tissue was incubated in a solution of Papain (BrainBits PAP) at 2 mg/mL for 30 min at 37 °C and dissociated in Hibernate-E for one minute using one sterile 9" silanized Pasteur pipette with a fire-polished tip. The cell dispersion solution was centrifuged at 214 g for 1 min, and the pellet was resuspended with 1 mL NbActiv1 (BrainBits NbActiv1 500 mL). Cell concentration was determined using a hemocytometer and neurons were plated in 12-well culture plates with 18-mm PDL-coated coverslips (Neuvitro Corporation GG-18-PDL) at a concentration of 1.3 million cells/well. Neurons were then incubated at 37 °C, 5% CO$_2$, performing half media changes every 3-4 days with fresh NbActiv1 supplemented with Primocin$^{TM}$ (InvivoGen ant-pm-1). Neurons infected with GCaMP6f as stated above were infected with AAV9-hSyn-Cre (Addgene #105553-AAV9) and AAV9-hSyn-TRPA1-myc-DIO (Salk GT3 core) at DIV4 and half media changes were performed the next day. Cultures were incubated at 37 °C, 5% CO$_2$ until DIV10-12 and were used in electrophysiology experiments.

**Rotarod**. Mouse locomotor behavior was evaluated on a Rotor-Rod (SD Instruments). Mice underwent a single day of training at a constant speed of 3 RPM to acclimate to the Rotor-Rod. The next day, mice were placed on a rod that started at 0 RPM and gradually increased to 30 RPM over a 5-minute period. The latency to fall off the rod was collected. Each mouse underwent 4 trials daily with a 20-minute inter-trial interval in which mice were returned to their cages. The latency to fall off was averaged across the three best trials. This procedure was repeated across 5 days. The experimenter was blinded as to the identity of groups.

**Sequences and annotations**. Ten TRPA1 peptide sequences were retrieved from the National Center for Biotechnology Information (NCBI) RefSeq database for human (*Homo sapiens*; NCBI Taxonomy 9606; RefSeq XP_016869435.1), mouse (*Mus musculus*; NCBI Taxonomy 10090; RefSeq NM_177781), beaver (*Castor canadensis*; NCBI Taxonomy 51338; RefSeq XP_020010675.1), alpaca (*Vicugna pacos*; NCBI Taxonomy 30538; RefSeq XP_006202494.1), donkey (*Equus asinus*; NCBI Taxonomy 9793; RefSeq XP_014709261.1), bat (*Eptesicus fuscus*; NCBI Taxonomy 29078; RefSeq XP_008148609.1), alligator (*Alligator mississippiensis*; NCBI Taxonomy 8496; RefSeq XP_006277080.1), snake (*Notechis scutatus*; NCBI Taxonomy 8663; RefSeq XP_026545023.1), molly (*Poecilia formosa*; NCBI Taxonomy 48698; RefSeq XP_007554661.1), and zebrafish (*Danio rerio*; NCBI Taxonomy 7955; RefSeq NP_001007066.1). The human TRPA1 sequence was also retrieved from the UniProtKB database and aligned to the human TRPA1 RefSeq sequence to confirm that the sequences were identical. Uniprot coordinates of major domains and features for human TRPA1 were used to annotate the sequence in Geneious Prime (version 2020.1.2).

**Ultrasound focal area and attenuation calculations**. The focal area was calculated based on determining the wavelength (λ) of ultrasound for various frequencies (f) using the equation λ = c/f (where c is the approximate speed of sound in brain tissue[83], 1500 m/s). The focal area is diffraction limited at approximately λ/

2. Brain tissue attenuates ultrasound at a much greater rate than water[83]. We estimated attenuation with the following equation: Attenuation = 0.8 dB/(cm x ultrasound frequency). The attenuation of the ultrasound signal at 5 mm depth in brain tissue is shown in Fig. 1a)

**Ultrasound pressure and temperature measurements**. Ultrasound pressure and temperature measurements were collected through ultrasound gel at the same position from the face of the lithium niobate transducer and within the brain tissue through the skull using a Precision Acoustics Fiber-Optic Hydrophone connected to a Tektronix TBS 1052B Oscilloscope and ThinkPad Ultrabook. To enable stereotaxic insertion into the brain, the Fiber-Optic Hydrophone probe was carefully threaded through a glass capillary allowing the tip to remain exposed. Cortical measurements were taken in ex vivo cranial tissue in which the jaw and palate were removed to expose the base of the brain. Using the center of the hypothalamus as coordinates 0,0,0, the hydrophone was inserted at AP + 1.2, ML 1.0 and lowered to a depth of -5.6 to approximate the location of the layer V motor cortex. The transducer was coupled to the skull via ultrasound gel and temperature and pressure measurements were collected. We used a similar setup to estimate the ultrasound pressure and temperature in our in vitro setups (calcium imaging and electrophysiology) about 50 μm away from the top of the glass coverslip (transducer is positioned below).

**Ultrasound transducer**. We used a set of custom-made single-crystalline 127.68 Y-rotated X-propagating lithium niobate transducers operating in the thickness mode, as described in[25]. The fundamental frequency was measured to be 7 MHz using non-contact laser Doppler vibrometry (Polytec, Waldbronn, Germany). The devices were diced to 12 mm × 12 mm and built in to the in vitro test setup. The transducers were coated with a conductive layer of Au with a thickness of 1 μm with 20 nm of Ti acting as an adhesion layer. A DC sputtering (Denton 635 DC Sputtering system) process was used to coat 4" wafers in an inert gas environment with a 2.3 mTorr pressure and rotation speed of 13 rpm, at a deposition rate of 1.5 A/s for Ti and 7 A/s for Au. Devices were diced to size using an automated dicing saw (DISCO 3220) and the resonance frequency verified using non-contact laser Doppler vibrometry. Custom transducers were fabricated with resonant frequencies of 1 MHz and 2 MHz to test ultrasound responsiveness of various channels at those stimuli (Boston Piezo-Optics, MA).

**Viruses**. pAAV.Syn.DIO.*hs*TRPA1-myc plasmid was custom-made by GenScript. synP.DIO.EGFP.WPRE.hGH was a gift from Ian Wickersham (Addgene viral prep # 100043-AAV9). pAAV.Syn.GCaMP6f.WPRE.SV40[26] was a gift from Douglas Kim & GENIE Project (Addgene viral prep # 100837-AAV9; http://n2t.net/addgene:100837; RRID:Addgene_100837). pENN.AAV.hSyn.Cre.WPRE.hGH was a gift from James M. Wilson (Addgene viral prep # 105553-AAV9; http://n2t.net/addgene:105553; RRID:Addgene_105553). AAV9-hsyn-DIO-*hs*TRPA1-myc (GT3 Core at Salk Institute of Biological Studies) was injected at either 4E13 along with 1E12 AAV9-hsyn-DIO-GFP (Addgene #100043-AAV9) diluted in Hank's Balance Salt Solution for injection. Adult male and female Npr3-cre mice (19-30 g) received 400nL unilateral injections to the right motor cortex at AP 0.0 ML -1.0, AP + 0.5 ML -1.0, AP + 0.5 ML-1.5 at DV 0.5[50]. Briefly, small holes were drilled (0.45 mm drill bit) into the skull over those coordinates, and virus was delivered through a pulled glass pipette at 2nL/sec by a Nanoject iii (Drummond Scientific Company). Successful viral delivery was confirmed post-mortem via immunohistochemistry for GFP and/or the myc-tag.

**Quantification and statistical analysis**. Statistical analyses were performed in GraphPad Prism and R. All statistical tests in this study were two-tailed. Single-variable comparisons were made with Mann-Whitney test. Group comparisons were made using either analysis of variance (ANOVA) followed by Tukey–Kramer post-hoc analysis or non-parametric Kruskal-Wallis test followed by Dunn's post-hoc analysis. The ROUT method in GraphPad Prism with a q = 0.2% was used to identify and exclude outliers. Statistics used to analyze calcium imaging data are described in Methods. No statistical methods were used to predetermine sample sizes for single experiments. All experiments and micrographs were repeated at least thrice in independent replicates unless indicated otherwise.

**Reporting summary**. Further information on research design is available in the Nature Research Reporting Summary linked to this article.

## Data availability

A source file with the datasets used in this study is also included. Publically available sequence data from National Center for Biotechnology Information (NCBI) RefSeq and UniProtKB were used in our study. Sequences include RefSeq IDs XP_016869435.1, NM_177781, XP_020010675.1, XP_006202494.1, XP_014709261.1, XP_008148609.1, XP_006277080.1, XP_026545023.1, XP_007554661.1, NP_001007066.1 and XM_017013946.1 and UniProtKB ID O75762. The reagents generated in this study, including pAAV-hSyn-DIO-hTRPA1-myc will be deposited in Addgene and are available upon reasonable request. Further information and requests for resources and

reagents should be directed to and will be fulfilled by the corresponding author. Source data are provided with this paper.

## Code availability

The code used to analyze calcium imaging data are available at https://github.com/shreklab/Duque-Lee-Kubli-Tufail2020.git.

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

## Acknowledgements

We thank Vivian Ko, Daniel Gibbs, Teresa Grider, Josh Nichols, Edward Callaway, Richard Daneman, Ken Diffenderfer and Mike Rieger for their technical help; Edward

Callaway, Marcos Sotomayor, Kathleen Quach, Javier How, and members of the Chalasani lab for their critical reading of the manuscript. This work was funded by grants from the National Institutes of Health (R01MH111534, R01NS115591), Brain Research Foundation, Innovation grants from Salk Institute and Kavli Institute of Brain and Mind (S.H.C.). C.L-K. is a Vertex Fellow of the Life Sciences Research Foundation. J.F. is grateful for funding from the W.M. Keck Foundation (SERF) in support of transducer design and fabrication for this work. This work was also supported by the Waitt Advanced Biophotonics and GT3 Cores with funding from NCI CCSG P30014195 and NINDSR24, respectively.

## Author contributions

M.D., C.L-K, designed and performed experiments, analyzed data and wrote the paper; Y.T., U.M., A.C., J.P., J.M.L., designed and performed experiments, and analyzed data; E.E. performed bioinformatic analysis; R.S. designed the membrane localization study and established the histology workflow; C.W. performed BaseScope analysis; A.V., J.F. designed, fabricated and validated ultrasound transducers; and S.H.C. designed experiments and wrote the paper.

## Competing interests

The authors declare no competing interests.
