## [Peer Review File · Nature Communications]

Reviewers' Comments:

Reviewer #1:

Remarks to the Author:

In this manuscript, the authors aim to explore an exogenous protein that can enhance the ultrasound sensitivity of genetically-modified cells. This would potentially solve the challenge that focused area of low-frequency ultrasound (~millimeter cubic) is too big to stimulate target neurons. After an unbiased screen of 191 mechanosensitive ion channels and other reported ultrasound-sensitive proteins, authors found that human TRPA1 outperforms other tested proteins regarding to its ultrasound sensitivity. A short pulse of 6.9 MHz ultrasound stimulation (100 ms) efficiently evokes calcium influx and membrane currents in human HEK cells and rodent cortical neurons as well as limb movements. Mechanismly, the N terminal tip region of human TRPA1, an unique domain that is only found in human but not in nonmammals' TRPA1s, is important for its ultrasound sensitivity. Moreover, the interactions between human TRPA1 and actin cytoskeleton and cholesterol are also critical for driving ultrasound-evoked hsTRPA1 responses. Overall, the current work is solid and gives new insights of how human TRPA1 senses ultrasound stimulation and triggers calcium response for neuromodulation. However, there are several concerns that need to be addressed before the manuscript can be considered for publication.

Major points:

1. According to the set-up of 2D cell culture (Fig. 1b), chambers and glass coverslips may attenuate the energy of ultrasound. It is necessary to measure the acoustic pressure of delivered ultrasound waves above glass coverslip.
2. The parameters of ultrasound (6.9 MHz, 100 ms, 2.5 MPa) temporally increase $\sim 1^\circ\text{C}$ (Extended Data Fig. S1a, b). The human TRPA1 is sensitive to temperature (Sinica et al., Cells, 2019). Moreover, N-terminal tip of TRPA1 affects its temperature sensitivity (Kang et al., Nature, 2011). It raises a possibility that the N-terminal tip of human TRPA1 might sense ultrasound-induced thermal effect and then contribute to the calcium responses in hsTRPA1-expressing cells. Authors need to address this assumption.
3. In Extended Data Fig. S2, there are few ROIs drawn in cells with non or low expression of dTomato which may lead to underestimation of hsTRPA1's response in these experiments. To avoid that, authors need to remove ROIs from cells with low dTomato intensity. Moreover, the size of hsTRPA1-transfected cells seems to be much larger than control cells (Extended Data Fig. S2a v.s. S2d). If hsTRPA1 does not significantly change the cell size, authors need to replace extended data Fig. S2a to a representative image.
4. It is not clear if authors tried to detect hsTRPA1 or mouse TRPA1 mRNA in mouse brains or AAV-infected brains. Moreover, which species of TRPA1 puncta signal was detected in adult WT DRG? Authors need to clearly describe the information and methods of this experiment.
5. Codons from non-human species significantly determine the exogenous expression efficiency in human cells and may consequently affect the ultrasound sensitivity of transfected cells. While the expression of dTomato serves as an indicator of transfected cells using 2A sequence, it can not fully represent the level of translated ion channels. Did authors apply codon optimization in their gene library? If so, authors must mention this information in the material and methods.

Minor points:

1. Line 1056, Remove "(,)" in a.
2. Line 1065, Remove ****p<0.0001 in d.
3. In Fig. 2c, the information of "Td", "Hg", "Cs", and "C.A." need to be described in figure legends.
4. Line 109, Extended Data Fig. 2c does not show what authors mentioned.
5. Line 110, Extended Data Fig. 1g, h does not show what authors mentioned.
6. Line 175, "Fig. 4a" needs to be corrected to "Fig. 4".
7. In extended Data Fig. S7, the information of scale bars is missing. Moreover, in S7e, "uM" need to be corrected to " μM ".
- 8, In extended data fig. S4, the information of scale bars is missing.
8. In Fig. 3e, "e" and the description of orange bars are missing.
9. In Fig. 3f, the description of "W.C." is missing.
11. In Fig. 5b, Fig. 6a, b, Extended Data Fig. S6, and Extended Data Fig. S10, " μM " need to be corrected to " μm ".

Reviewer #2:

Remarks to the Author:

The authors present a very interesting and comprehensive work on identifying the Transient Receptor Potential A1 (hsTRPA1) channels as a highly efficient sonogenetic protein sensitive to a type of ultrasound stimulation. The efficiency and mechanisms were tested and validated through GCaMP6f fluorescence and patch-clamp electrophysiological quantifications on in vitro human embryonic kidney-293T (HEK) cells and rodent primary neurons using chemical agonist and antagonist of TRPA1, as well as actin/microtubule depolymerizer and stabilizer drugs. The ultrasound sensitivity of hsTRPA1 was further investigated on in vivo transgenic mice neurons through EMG and movement measurements. The presented c-Fos quantification in auditory cortex would also be useful to rule out the auditory side effects in the ultrasound neuromodulation. Overall, this innovative work is solid and impactful to the brain stimulation field. However, the authors may need to address the following questions carefully.

Major issues:

1. It would be necessary to include spatial characterizations of the produced ultrasound pressure/intensity field by the custom-made lithium niobate transducer, especially the transcranial ultrasound field in the in vivo experiments. In the in vitro experimental setup, what is the acoustic attenuation through the glass coverslip? As the ultrasound has a normal incidence, it would also be necessary to characterize the ultrasound pressure field distribution within the cell media and see if any standing wave pattern was induced or not, which might change the reported ultrasound pressure level.
2. The investigations on the putative mechanisms of hsTRPA1's sensitivity to ultrasound stimulation is interesting. The authors reached the conclusion that "actin depolymerization by cytochalasin D treatment selectively blocked hsTRPA1 responses to ultrasound, but not chemical agonist, demonstrating a specific role for the actin cytoskeleton in ultrasound sensation" (lines 191-192), it is not clear that how they ruled out the latrunculin A, as both drugs are actin depolymerization agents and both can lead to reductions of ultrasound responses. In addition, please clarify what the vehicle is.
3. In lines 204-205, regarding the statement that "This mutation caused a complete loss of ultrasound sensitivity without affecting responsiveness to AITC...", somehow the Fig. 3j shows a significantly enhanced responsiveness to the agonist. Please check and see whether this claim needs to be corrected based on the data.
4. In lines 291-294, were the negative peak pressures ranging from 0.35 – 1.05 MPa measured before or after the skull? In the extended Fig. 11, there are two cases with ultrasound pressures over 1.5 MPa. Also, it seems that ultrasound pressures are higher at deeper brain than those shallower locations. How were the ultrasound pressure levels controlled along the depth in midbrain and hindbrain using the flat custom-made transducer?
5. In lines 392-393, the authors mentioned that "responses in control neurons in vitro could be likely an artefact..." originated from several potential sources. It would be nontrivial to validate that the recording from the hsTRPA1-expressing neurons will be artefacts free, and this would be a critical issue when MPa ultrasound was applied.
6. In lines 552-553, the ketamine+xylazine anesthesia was adopted in the EMG experiments. How did the authors control the anesthesia level for inducing non-biased assessments of limb movements as presented in Fig. 5d?

Minor issues:

1. Are the contents presented in lines 15-26 considered as Abstract?
2. In line 29, please add "fundamental" before "frequencies" to make this statement clear.

3. In line 52, how was the "acceptable loss in energy" was supported? I suspect the 6.91 MHz ultrasound may have large acoustic insertion loss especially for the transcranial application.
4. In line 71, comparing the stated numbers and those presented in Fig. 1a, I would suggest to change the labeled "7 MHz" to "6.91 MHz" to be consistent, as the focal volumes use the same number.
5. In line 81, "Extended Data Fig. 1-m" should be "Extended Data Fig. 1d-m". Please check.
6. In line 100-101, it is good to confirm that "hsTRPA1 channel was sensitive to a broad range of ultrasound stimuli". However, the much smaller focal volume of 6.91 MHz, comparing to those of 1 and 1.978 MHz, may also raise another practical challenge in aligning the ultrasound focus with the electrophysiological recording probes. How did the authors make sure this alignment would be good enough for a reliable electrical recording?
7. In line 91 and line 111, the chemical agonist AITC were defined twice.
8. In line 124, what is "more numerous"?
9. In line 232, "Fig. 3e" should be "Fig. 4e", and the panel e of Fig. 4 need to be labelled.
10. In line 261, change "compared to" to "than".
11. In line 267-268, the control neurons did not present action potentials, however, in an earlier work (Tufail et al. Neuron 2010), an even lower ultrasound pressure was able to elicit action potentials in wild-type mice as well as movement. What could be the major reason? Would a different fundamental frequency lead to such a big difference?
12. In line 338, change "single pulses" to "pulses".
13. In line 382, what is "MCD"?
14. In line 1056, remove the unnecessary brackets for the caption texts of Fig. 1a.
15. In lines 1057-1058, change "Plot showing c, ..." to "c, Plot showing ...".
16. In the Fig. 2a, change "TRPA1" to "hsTRPA1" to be more specific and change "dtom" to "dTom" to make the naming convention for dTom consistent.
17. In line 1085, please include the full name of DIC here. Is the image generated by Differential Interference Contrast microscopy?
18. In lines 1095-1096, the significance levels are described in the caption for panel e, but none of those labels are used in panel e.
19. In the caption texts of Fig. 5, please add brief statistical details, such as the significance levels and whether any correction method was applied in the multiple comparisons.

Reviewer #3:

Remarks to the Author:

In the present study by Duque and colleagues, the human TRPA1 (hsTRPA1) is selected out of a screen of 191 candidate channels as a superior mechanosensitive sonogenetic tool conferring ultrasound sensitivity to mammalian cells in vitro as well as in vivo. This study is interesting as it could help to develop a broadly usable sonogenetic tool to explore cellular communication in many species. The study is ambitious as it also addresses the mechanism by which TRPA1 displays species specific mechanosensory properties, something that needs further in-depth investigations. Please find below my comments and points for consideration:

hsTRPA1 is a sonogenetic candidate

Even though HEK cells stably expressing GCaMP6f were used, did the authors confirm that cells with no or little calcium responses still responded to calcium ionophore?

Line 77; 1.5 MPa or 2.5 MPa as in Fig. 1e?

Line 81; Fig. 1-m?

Line 89; should read (Extended Data Fig. 1c and 2b)?

Line 109; should read (Extended Data Fig. 1c and 2b)?

Line 110; should read (Extended Data Fig. 1c and 2e)?

Fig. 1, legend line 1064; N = 3 repeated measurements on the same coverslip/transfection, or is it N = 1 in triplicate? Please clarify.

Fig.2, legend line 1081; define right bottom image in (a).

Line 117; the cell-attached configuration was used in electrophysiological experiments on HEK cells. The same US device setup was used as in imaging experiments. This means that hsTRPA1 in the patch was indirectly exposed to US pressure (0.15 MPa) in contrast to in imaging experiments (2.5 MPa)? Have the authors tried to activate hsTRPA1 within the patch by changing the pipette pressure, which has been commonly used in studies identifying mechanosensitive proteins? In Fig. 2d and e, how was hsTRPA1 expression confirmed functionally? Could it be that presumed hsTRPA1 patches displayed non hsTRPA1 activity? What is N and was the recording time standardized? What is the activity in WT? Is it not present in preparations exposed to US as in Fig. 2f?

Line 125; the US current magnitude was of similar magnitude as in ref 39. Please specify as these authors used different conditions (patch-clamp configuration, ion composition etc.), and TRPA1 channel current properties differ hugely depending on experimental conditions and stimuli (Zygmunt & Högestätt *Handb Exp Pharmacol* 2014, Table 1).

Line 127; peak amplitudes, not events, were inhibited by HC030031 (Fig. 2i). What was the effect of HC030031 on events?

Fig. 2, line 1091; relative peak amplitude (I/I_{max}). What is I_{max} ? In responses to AITC or NMM?

Putative mechanisms underlying ultrasound sensitivity of TRPA1

Line 144; update ref 42 and add Suo et al *Neuron* 105, 882-894, 2020.

Although many studies have shown an involvement of TRPA1 in mammalian mechanosensation including nociception (Zygmunt & Högestätt *Handb Exp Pharmacol* 2014, Table 4; Talavera et al *Physiol Rev* 2020), the role of TRPA1 as a mechanosensor itself has been disputed and only recently demonstrated (Moparthi and Zygmunt, *Cell Calcium* 2020). The present study adds further support that mammalian TRPA1s respond to pressure. Naturally, the molecular mechanism by which pressure is converted into channel activity still remains to be determined. Here, the authors conclude that the N-terminal tip is critical to respond to pressure.

Line 160; only a full concentration-response relationship can reveal a change in sensitivity, not a single concentration of AITC (Fig. 3c). Likewise, the sensitivity of N-tip a.m to AITC may not be different from WT although the response was smaller, and the N-tip d.r may be less sensitive although a larger response (Fig. 3e). Also, truncated and chimeras may still respond at higher pressures than 2.5 MPa. Nevertheless, under the chosen experimental conditions it seems reasonable to conclude that the structural changes of the N-tip region affect the hsTRPA1 US sensitivity. Interestingly, the N-terminal ARD is not needed for hsTRPA1 intrinsic mechanosensitivity (Moparthi and Zygmunt *Cell Calcium* 2020) although it most likely contributes/modulates the mechanosensory properties of TRPA1 in a cellular context by interaction with the cell membrane and/or cytoskeleton, as also proposed in the present study. Also, it is important to note that species differences in mechanosensation could be related to the cellular environment influencing TRPA1 conformation and as a result a change in mechanosensitivity. Thus, further in-depth studies including other means of pressure activation are needed to fully understand the evolutionary TRPA1 mechanosensory properties.

The role of cholesterol and lipids in the regulation of TRP channels are indeed very interesting and the authors find a similar important functional interaction between cholesterol and TM2 as previously shown in mouse TRPA1 (Startek et al *eLife* 2019). However, in contrast to the mouse TRPA1 treated with MCD, E_{max} response may not be reduced in MCD treated hsTRPA1. However, full concentration-response curves are needed to disclose any change in hsTRPA1 sensitivity as well as E_{max} to AITC as is the case for mouse TRPA1 (Startek et al *eLife* 2019).

hsTRPA1 potentiates ultrasound-responses in primary neurons

Fig. 4, line 1157; control plasmids?

Fig. 4; number of experiments in d and e?

Fig. 4, line 1161; clearly state that electrophysiology experiments were performed with rat neurons.

Line 232; 3e should be 4e and e is missing in Fig. 4.

Extended Data Fig. S7; legend is unclear and not complete for d-f. (a) Concentration response (Dose when administered in vivo).

What is the hsTRPA1 expression efficiency in rat neurons? Was the presence of hsTRPA1 confirmed with inhibitors also in the electrophysiology experiments? What is the current in Cre-only controls? Even though this current may not evoke action potentials as shown in Fig. 4 (j,k), it could still release signaling molecules from these neurons and evoke physiological responses thought to be hsTRPA1 mediated in US stimulated hsTRPA1 sonogenetic experiments. What is the number of events in experiments shown in Fig. 4g,h. This could be as relevant as amplitude (Fig. 4i) to report? Please define time frame of these experiments and those in Fig. 4j,k. Also, define pulse duration. Could action potentials be repeated? Number of experiments, transfections, preparations, animals etc., is unclear here and in most experiments/figure legends throughout the study.

Extended Data Fig. S9; no significance is shown in e.

hsTRPA1 confers ultrasound sensitivity in vivo

Fig. 5; statistical analysis in legend? What is the number of animals used?

Have the authors considered to express the hsTRPA1/N-tip d.r. chimera as a negative control (as well as in primary neurons)?

Is it possible that hsTRPA1 is activated by free radicals formed by US stimulation in vivo as well as in vitro, and that species differences observed in the present study are influenced by TRPA1 channel species-dependent redox sensitivity?

DISCUSSION

Line 358; hTRPA1 is intrinsically mechanosensitive (Moparthy and Zygmunt Cell Calcium 2020).

Line 372; Mosquito and hTRPA1 are thermo- and mechano-sensitive also without the N-ARD (Survery et al JBC 2016; Moparthy et al PNAS 2014; Moparthy and Zygmunt Cell Calcium 2020).

Line 382; MCD reduced both sensitivity and Emax responses to AITC in mTRPA1.

Point-by-point response (in black) to each of the reviewer's comments (blue).

Reviewer #1

In this manuscript, the authors aim to explore an exogenous protein that can enhance the ultrasound sensitivity of genetically-modified cells. This would potentially solve the challenge that focused area of low-frequency ultrasound (~millimeter cubic) is too big to stimulate target neurons. After an unbiased screen of 191 mechanosensitive ion channels and other reported ultrasound-sensitive proteins, authors found that human TRPA1 outperforms other tested proteins regarding to its ultrasound sensitivity. A short pulse of 6.9 MHz ultrasound stimulation (100 ms) efficiently evokes calcium influx and membrane currents in human HEK cells and rodent cortical neurons as well as limb movements. Mechanismly, the N terminal tip region of human TRPA1, an unique domain that is only found in human but not in nonmammals' TRPA1s, is important for its ultrasound sensitivity. Moreover, the interactions between human TRPA1 and actin cytoskeleton and cholesterol are also critical for driving ultrasound-evoked hsTRPA1 responses.

Overall, the current work is solid and gives new insights of how human TRPA1 senses ultrasound stimulation and triggers calcium response for neuromodulation. However, there are several concerns that need to be addressed before the manuscript can be considered for publication.

We thank the reviewer for their positive feedback.

Major points:

1. According to the set-up of 2D cell culture (Fig. 1b), chambers and glass coverslips may attenuate the energy of ultrasound. It is necessary to measure the acoustic pressure of delivered ultrasound waves above glass coverslip.

Yes, we have measured the ultrasound energy above the coverslip and have included that information in the revised manuscript.

2. The parameters of ultrasound (6.9 MHz, 100 ms, 2.5 MPa) temporally increase ~1°C (Extended Data Fig. S1a, b). The human TRPA1 is sensitive to temperature (Sinica et al., Cells, 2019). Moreover, N-terminal tip of TRPA1 affects its temperature sensitivity (Kang et al., Nature, 2011). It raises a possibility that the N-terminal tip of human TRPA1 might sense ultrasound-induced thermal effect and then contribute to the calcium responses in hsTRPA1-expressing cells. Authors need to address this assumption.

Human TRPA1 is also considered to be sensitive to cold temperatures (<https://www.pnas.org/content/111/47/16901>), which does not require its N-terminal tip domain. In addition, the temperature response might also be dependent on reactive oxygen species (<https://www.nature.com/articles/ncomms12840>). It is however possible that hsTRPA1 is responding to a quick < 0.4°C rise in temperature in our ultrasound-imaging assay (which provides greater sensitivity and faster kinetics than the previously

reported temperature sensors). Also, we do not believe that the ultrasound effect on HEK cells is driven by temperature changes as a known temperature sensitive channel TRPV1 is not effective in triggering calcium changes (Extended data figure S2I). We have included this in our revised discussion.

3. In Extended Data Fig. S2, there are few ROIs drawn in cells with non or low expression of dTomato which may lead to underestimation of hsTRPA1's response in these experiments. To avoid that, authors need to remove ROIs from cells with low dTomato intensity. Moreover, the size of hsTRPA1-transfected cells seems to be much larger than control cells (Extended Data Fig. S2a v.s. S2d). If hsTRPA1 does not significantly change the cell size, authors need to replace extended data Fig. S2a to a representative image.

These ROIs were drawn using an automated Fiji script (see Methods), which sometimes include cells with low dTomato expression, in both control and hsTRPA1 groups. In general, dTomato appears brighter in control cells, since the TRPA1 construct required cleavage of p2A to achieve proper dTOM trafficking. We apologize for the confusion, and have replaced the representative images in Extended data fig S2a since hsTRPA1 expression had no correlation with the cell size.

4. It is not clear if authors tried to detect hsTRPA1 or mouse TRPA1 mRNA in mouse brains or AAV-infected brains. Moreover, which species of TRPA1 puncta signal was detected in adult WT DRG? Authors need to clearly describe the information and methods of this experiment.

We used an ACD Bio BaseScope probe designed to target mouse TRPA1 mRNA in mouse brain to detect native expression of mouse TRPA1 in wild-type and TRPA1 knockout mice. We have included this information in the revised manuscript.

5. Codons from non-human species significantly determine the exogenous expression efficiency in human cells and may consequently affect the ultrasound sensitivity of transfected cells. While the expression of dTomato serves as an indicator of transfected cells using 2A sequence, it can not fully represent the level of translated ion channels. Did authors apply codon optimization in their gene library? If so, authors must mention this information in the material and methods.

All constructs in the library were codon optimized for expression in human cells. We will include this information in the revised material and methods. Because we do not expect the dTomato expression to accurately reflect the expression levels of the translated ion channels, we use dTomato expression as a yes/no criterion for inclusion in calcium analysis regardless of dTomato expression intensity. This is included in the revised methods

Minor points:

1. Line 1056, Remove “(“,”)” in a.
2. Line 1065, Remove ****p<0.0001 in d.
3. In Fig. 2c, the information of “Td”, “Hg”, “Cs”, and “C.A.” need to be described in figure legends.
4. Line 109, Extended Data Fig. 2c does not show what authors mentioned.
5. Line 110, Extended Data Fig. 1g, h does not show what authors mentioned.
6. Line 175, “Fig. 4a” needs to be corrected to “Fig. 4”.
7. In extended Data Fig. S7, the information of scale bars is missing. Moreover, in S7e, “uM” need to be corrected to “ μ M”.
8. In extended data fig. S4, the information of scale bars is missing.
8. In Fig. 3e, “e” and the description of orange bars are missing.
9. In Fig. 3f, the description of “W.C.” is missing.
11. In Fig. 5b, Fig. 6a, b, Extended Data Fig. S6, and Extended Data Fig. S10, μ ”M” need to be corrected to μ ”m”.

All of these are fixed in the revised manuscript.

Reviewer #2 (Remarks to the Author):

The authors present a very interesting and comprehensive work on identifying the Transient Receptor Potential A1 (hsTRPA1) channels as a highly efficient sonogenetic protein sensitive to a type of ultrasound stimulation. The efficiency and mechanisms were tested and validated through GCaMP6f fluorescence and patch-clamp electrophysiological quantifications on in vitro human embryonic kidney-293T (HEK) cells and rodent primary neurons using chemical agonist and antagonist of TRPA1, as well as actin/microtubule depolymerizer and stabilizer drugs. The ultrasound sensitivity of hsTRPA1 was further investigated on in vivo transgenic mice neurons through EMG and movement measurements. The presented c-Fos quantification in auditory cortex would also be useful to rule out the auditory side effects in the ultrasound neuromodulation. Overall, this innovative work is solid and impactful to the brain stimulation field. However, the authors may need to address the following questions carefully.

We thank the reviewer for their positive feedback.

Major issues:

1. It would be necessary to include spatial characterizations of the produced ultrasound pressure/intensity field by the custom-made lithium niobate transducer, especially the transcranial ultrasound field in the in vivo experiments. In the in vitro experimental setup, what is the acoustic attenuation through the glass coverslip? As the ultrasound has a normal incidence, it would also be necessary to characterize the ultrasound pressure field distribution within the cell media and see if any standing wave pattern was induced

or not, which might change the reported ultrasound pressure level.

We have included data about attenuation through the glass coverslip in our revised manuscript. The ultrasound pressure levels reported in figure S1 are measured as close to the coverslip as possible without damaging our fiber optic hydrophone. This ensures that we capture pressure readings on top of the coverslip and as close as possible, about 50 μm away from the cultured and imaged cells on top of the coverslip.

2. The investigations on the putative mechanisms of hsTRPA1's sensitivity to ultrasound stimulation is interesting. The authors reached the conclusion that "actin depolymerization by cytochalasin D treatment selectively blocked hsTRPA1 responses to ultrasound, but not chemical agonist, demonstrating a specific role for the actin cytoskeleton in ultrasound sensation" (lines 191-192), it is not clear that how they ruled out the latrunculin A, as both drugs are actin depolymerization agents and both can lead to reductions of ultrasound responses. In addition, please clarify what the vehicle is.

The vehicle is DMSO. We apologize about the misunderstanding regarding latrunculin A. Even though they both do have a significant effect in ultrasound responses, latrunculin A also significantly reduces the response to a chemical agonist AITC (Extended Data Fig 5c). Thus, we are unable to separate whether latrunculin A acts specifically on TRPA1s ultrasound sensitivity, or it just generally decreases cell health or interferes with channel general function. However, since cytochalasin-D seems to specifically interfere with the ultrasound sensitivity, we conclude actin cytoskeleton might be involved in the sensitivity of the channel to ultrasound. Moreover, latrunculin A and cytochalasin D use different mechanisms to affect actin ([https://www.cell.com/biophysj/fulltext/S0006-3495\(00\)76614-8](https://www.cell.com/biophysj/fulltext/S0006-3495(00)76614-8)). We have clarified this in the revised manuscript.

3. In lines 204-205, regarding the statement that "This mutation caused a complete loss of ultrasound sensitivity without affecting responsiveness to AITC...", somehow the Fig. 3j shows a significantly enhanced responsiveness to the agonist. Please check and see whether this claim needs to be corrected based on the data.

We have corrected the statement in the revised manuscript (meant to say that it does not decrease responsiveness to AITC).

4. In lines 291-294, were the negative peak pressures ranging from 0.35 – 1.05 MPa measured before or after the skull? In the extended Fig. 11, there are two cases with ultrasound pressures over 1.5 MPa. Also, it seems that ultrasound pressures are higher at deeper brain than those shallower locations. How were the ultrasound pressure levels controlled along the depth in midbrain and hindbrain using the flat custom-made transducer?

In lines 291-294, the negative peak pressures ranging from 0.35-1.05MPa were measured inside of the mouse skull with the brain intact; the same is true for the measurements in extended Fig 11. While we did find pressures at deeper brain regions that were higher than 1.5MPa, these brain regions were outside the scope of any experiments performed *in vivo*, and

could be a result of the relationship between the distance from the transducer face, the shape of the mouse skull, reflection of the acoustic wave within, as well as other factors that we would explore in future studies. As such, we did not control for pressure along the depth in midbrain and hindbrain but rather characterized that ultrasound does propagate into deeper brain regions to highlight the benefit of using penetrance of ultrasound versus lower penetrance of light as a stimulation modality. We have begun to make inroads to solving this problem using a diffuser attached to the transducer (<https://www.biorxiv.org/content/10.1101/2021.08.21.457135v1>).

5. In lines 392-393, the authors mentioned that “responses in control neurons in vitro could be likely an artefact...” originated from several potential sources. It would be nontrivial to validate that the recording from the hsTRPA1-expressing neurons will be artefacts free, and this would be a critical issue when MPa ultrasound was applied.

We apologize for the use of the word “artefact”. Our calcium imaging data shows that control neuron responses to ultrasound are blocked by tetrodotoxin treatment, indicating a role for voltage-gated sodium channels. Also, we recently published a study in *Advanced Science* showing that ultrasound deflects cellular membranes, which in turn changes membrane capacitance and generates an action potential. While we don’t know if this mechanism is also used in the sonogenetic context, but have included this in our revised manuscript.

6. In lines 552-553, the ketamine+xylazine anesthesia was adopted in the EMG experiments. How did the authors control the anesthesia level for inducing non-biased assessments of limb movements as presented in Fig. 5d?

The dose of ketamine xylazine anesthesia was applied in a consistent weight-dependent fashion and the depth of anesthesia was monitored throughout the recording session to ensure that mice did not respond to toe pinch. In cases of recording sessions in which the mice began to respond to toe pinch, supplemental half doses of ketamine/xylazine was administered as needed.

Minor issues:

1. Are the contents presented in lines 15-26 considered as Abstract?

Yes

2. In line 29, please add “fundamental” before “frequencies” to make this statement clear.

Fixed

3. In line 52, how was the “acceptable loss in energy” was supported? I suspect the 6.91 MHz ultrasound may have large acoustic insertion loss especially for the transcranial application.

We do lose about 75% of the energy delivered through the glass coverslip or the mouse skull. We have included this information in the revised manuscript. While this is a large loss, we feel that the ability of this transducer to still transmit sufficient energy into the

brain for our purposes combined with the small focal volume of this transducer gives us an advantage for rodent studies.

4. In line 71, comparing the stated numbers and those presented in Fig. 1a, I would suggest to change the labeled “7 MHz” to “6.91 MHz” to be consistent, as the focal volumes use the same number.

Fixed throughout the manuscript

5. In line 81, “Extended Data Fig. 1-m” should be “Extended Data Fig. 1d-m”. Please check.

Fixed

6. In line 100-101, it is good to confirm that “hsTRPA1 channel was sensitive to a broad range of ultrasound stimuli”. However, the much smaller focal volume of 6.91 MHz, comparing to those of 1 and 1.978 MHz, may also raise another practical challenge in aligning the ultrasound focus with the electrophysiological recording probes. How did the authors make sure this alignment would be good enough for a reliable electrical recording?

Our imaging and electrophysiological recordings used the same 6.91MHz Lithium Niobate transducer. Due to technical reasons, we were only able to reliably use 6.91MHz for in-vitro patch recordings. For this reason, we chose to culture neurons on 18mm round #1 glass coverslips, knowing that the transducer size covered majority of the cultured area. This allowed us to visually search for morphologically healthy neurons that were located directly over the transducer. Another significant technical factor that influenced the quality of electrophysiology experiments was placing the transducer at a defined distance away from the cultured neurons to maintain consistent US delivery. This was accomplished by leveling the transducer below the recording chamber, aligned with a pedestal tower connected to a rigid sliding dovetail bridge. We have described this in our revised methods.

7. In line 91 and line 111, the chemical agonist AITC were defined twice.

Fixed

8. In line 124, what is “more numerous”?

Inward currents. Fixed

9. In line 232, “Fig. 3e” should be “Fig. 4e”, and the panel e of Fig. 4 need to be labelled. fixed

10. In line 261, change “compared to” to “than”.

Fixed

11. In line 267-268, the control neurons did not present action potentials, however, in an earlier work (Tufail et al. Neuron 2010), an even lower ultrasound pressure was able to elicit action potentials in wild-type mice as well as movement. What could be the major reason? Would a different fundamental frequency lead to such a big difference?

There are two main factors that differentiate Tufail et al. Neuron 2010 and our current study. Firstly, the fundamental frequency of our current study is significantly different than those used in Tufail et al. 2010 (eg. 0.5MHz vs 6.91MHz). One major experimental advantage this gave us was the ability to better record electrical properties of neurons using the patch-clamp technique. Previously (Tyler et al. 2008), we had very limited success using patch approaches as the lower US frequencies more readily resonated with the patch pipette, resulting with frequent loss of whole-cell seals. Secondly, our current study did not require the use of a pulse-repetition frequency (PRF) to modulate the waveform. It has been recently demonstrated (Kai Yu et al. Nature Communications 2021) that utilizing a PRF throughout the stimulus waveform produces drastic differences in evoked activity of neuronal sub-populations. Lastly, it is difficult to directly compare stimulation parameters when implementing in-vitro or in-vivo recording techniques. We believe that our approach demonstrates an example where the US parameter space is more readily comparable when translating from in vitro to in vivo experimental designs. We have included this in our revised discussion

12. In line 338, change “single pulses” to “pulses”.

Fixed

13. In line 382, what is “MCD”?

Fixed. methyl- β -cyclodextrin

14. In line 1056, remove the unnecessary brackets for the caption texts of Fig. 1a.

Fixed

15. In lines 1057-1058, change “Plot showing c, ...” to “c, Plot showing ...”.

Fixed

16. In the Fig. 2a, change “TRPA1” to “hsTRPA1” to be more specific and change “dtom” to “dTom” to make the naming convention for dTom consistent.

Fixed

17. In line 1085, please include the full name of DIC here. Is the image generated by Differential Interference Contrast microscopy?

Yes, we used DIC to mean differential interference contrast microscopy (DIC). Kohler illumination was initially established to allow for proper DIC images.

18. In lines 1095-1096, the significance levels are described in the caption for panel e, but none of those labels are used in panel e.

Fixed

19. In the caption texts of Fig. 5, please add brief statistical details, such as the significance levels and whether any correction method was applied in the multiple comparisons.

Fixed

Reviewer #3

In the present study by Duque and colleagues, the human TRPA1 (hsTRPA1) is selected out of a screen of 191 candidate channels as a superior mechanosensitive sonogenetic tool conferring ultrasound sensitivity to mammalian cells in vitro as well as in vivo. This study is interesting as it could help to develop a broadly usable sonogenetic tool to explore cellular communication in many species. The study is ambitious as it also addresses the mechanism by which TRPA1 displays species specific mechanosensory properties, something that needs further in-depth investigations. Please find below my comments and points for consideration:

hsTRPA1 is a sonogenetic candidate

Even though HEK cells stably expressing GCaMP6f were used, did the authors confirm that cells with no or little calcium responses still responded to calcium ionophore?

We have included this data in the revised manuscript.

Line 77; 1.5 MPa or 2.5 MPa as in Fig. 1e?

2.5 MPa. We used 1.5 MPa for the screen, but we identified a candidate used 2.5 MPa for the rest of our in vitro studies. Clarified in the manuscript.

Line 81; Fig. 1-m?

Fixed. 1d-m

Line 89; should read (Extended Data Fig. 1c and 2b)?

Fixed.

Line 109; should read (Extended Data Fig. 1c and 2b)?

Fixed.

Line 110; should read (Extended Data Fig. 1c and 2e)?

Fixed.

Fig. 1, legend line 1064; N = 3 repeated measurements on the same coverslip/transfection, or is it N = 1 in triplicate? Please clarify.

Clarified

Fig.2, legend line 1081; define right bottom image in (a).

fixed

Line 117; the cell-attached configuration was used in electrophysiological experiments on HEK cells. The same US device setup was used as in imaging experiments. This means that hsTRPA1 in the patch was indirectly exposed to US pressure (0.15 MPa) in contrast to in imaging experiments (2.5 MPa)? Have the authors tried to activate hsTRPA1 within the patch by changing the pipette pressure, which has been commonly used in studies identifying mechanosensitive proteins? In Fig. 2d and e, how was

hsTRPA1 expression confirmed functionally? Could it be that presumed hsTRPA1 patches displayed non hsTRPA1 activity? What is N and was the recording time standardized? What is the activity in WT? Is it not present in preparations exposed to US as in Fig. 2f?

We did not specifically investigate the sensitivity of TRPA1 transduced cells to slow, low negative membrane pressures induced via patch pipette suction. In HEK cells, TRPA1 expressing cells did exhibit a modest increase in spontaneous channel activity compared to controls (figure 2D). In contrast, when recording from primary neurons, spontaneous action potential activity was not observed to be different to a reasonable qualitative extent between AAV-hsTRPA1 and AAV-Cre controls. It is likely we did not see hsTRPA1 dependent activation due to patch-pipette suction because hsTRPA1 transduced neurons tended to robustly respond to 6.91MHz and we focused on monitoring action-potential activity rather than channel kinetics in primary neurons. The recording time was standardized for each recording session. When suitable membrane seals were maintained, each gap-free recording trial lasted 5 minutes, unless membrane resistance stability was lost. Data from at least 8 cells are included in the manuscript. These details are included in the revised manuscript after line 335.

Line 125; the US current magnitude was of similar magnitude as in ref 39. Please specify as these authors used different conditions (patch-clamp configuration, ion composition etc.), and TRPA1 channel current properties differ hugely depending on experimental conditions and stimuli (Zygmunt & Högestätt Handb Exp Pharmacol 2014, Table 1).

Have included this information in the revised methods.

Line 127; peak amplitudes, not events, were inhibited by HC030031 (Fig. 2i). What was the effect of HC030031 on events?

Data included in Figure 2J.

Fig. 2, line 1091; relative peak amplitude (I/I_{max}). What is I_{max} ? In responses to AITC or NMM?

I/I_{max} is computed from the steady state current response after a depolarizing voltage step (stepping from -120mV to +120mV). I_{max} represents the largest current response obtained from a voltage step for each recorded cell. Details are included in the methods section.

Putative mechanisms underlying ultrasound sensitivity of TRPA1

Line 144; update ref 42 and add Suo et al Neuron 105, 882-894, 2020.

Although many studies have shown an involvement of TRPA1 in mammalian mechanosensation including nociception (Zygmunt & Högestätt Handb Exp Pharmacol

2014, Table 4; Talavera et al *Physiol Rev* 2020), the role of TRPA1 as a mechanosensor itself has been disputed and only recently demonstrated (Moparthy and Zygmunt, *Cell Calcium* 2020). The present study adds further support that mammalian TRPA1s respond to pressure. Naturally, the molecular mechanism by which pressure is converted into channel activity still remains to be determined. Here, the authors conclude that the N-terminal tip is critical to respond to pressure.

Line 160; only a full concentration-response relationship can reveal a change in sensitivity, not a single concentration of AITC (Fig. 3c). Likewise, the sensitivity of N-tip a.m to AITC may not be different from WT although the response was smaller, and the N-tip d.r may be less sensitive although a larger response (Fig. 3e). Also, truncated and chimeras may still respond at higher pressures than 2.5 MPa. Nevertheless, under the chosen experimental conditions it seems reasonable to conclude that the structural changes of the N-tip region affect the hsTRPA1 US sensitivity. Interestingly, the N-terminal ARD is not needed for hsTRPA1 intrinsic mechanosensitivity (Moparthy and Zygmunt *Cell Calcium* 2020) although it most likely contributes/modulates the mechanosensory properties of TRPA1 in a cellular context by interaction with the cell membrane and/or cytoskeleton, as also proposed in the present study. Also, it is important to note that species differences in mechanosensation could be related to the cellular environment influencing TRPA1 conformation and as a result a change in mechanosensitivity. Thus, further in-depth studies including other means of pressure activation are needed to fully understand the evolutionary TRPA1 mechanosensory properties.

We agree with the reviewer. In order to reveal the precise molecular mechanisms underlying ultrasound/pressure sensitivity in hsTRPA1, further studies including dose-response curves for different treatments and chimeras, are needed. However, our intention was to narrow down which region of the hsTRPA1 protein and identify the cellular elements that are required for ultrasound sensitivity as opposed to a chemical ligand. Our intent is to further understand the mechanism and lead to rational design of ultrasound-sensitive channels. We hope the reviewer agrees that our single-dose experiments achieve that goal. We also agree with the reviewer that the n-tip domain might have additional roles in the cell and have included a comment about this in our revised manuscript.

The role of cholesterol and lipids in the regulation of TRP channels are indeed very interesting and the authors find a similar important functional interaction between cholesterol and TM2 as previously shown in mouse TRPA1 (Startek et al *eLife* 2019). However, in contrast to the mouse TRPA1 treated with MCD, Emax response may not be reduced in MCD treated hsTRPA1. However, full concentration-response curves are needed to disclose any change in hsTRPA1 sensitivity as well as Emax to AITC as is the case for mouse TRPA1 (Startek et al *eLife* 2019).

We agree that in order to compare the effect of MCD on Emax to AITC between mouse and human homologs a full concentration-response curve is needed. We have

additional concentrations of AITC and compare the effects of MCD on HEK calcium responses. Data is included in S5e

hsTRPA1 potentiates ultrasound-responses in primary neurons

Fig. 4, line 1157; control plasmids?
Cre plasmids. Fixed

Fig. 4; number of experiments in d and e?
Included

Fig. 4, line 1161; clearly state that electrophysiology experiments were performed with rat neurons.
Included

Line 232; 3e should be 4e and e is missing in Fig. 4.
Fixed

Extended Data Fig. S7; legend is unclear and not complete for d-f. (a) Concentration response (Dose when administered in vivo).
Fixed

What is the hsTRPA1 expression efficiency in rat neurons?

TRPA1 expression was confirmed with quantitative PCR (qPCR) and was similar to the expression observed for mouse neurons. We have added a supplementary figure showing TRPA1 expression in rat cultures (Fig.S9 j,k).

Was the presence of hsTRPA1 confirmed with inhibitors also in the electrophysiology experiments? What is the current in Cre-only controls? Even though this current may not evoke action potentials as shown in Fig. 4 (j,k), it could still release signaling molecules from these neurons and evoke physiological responses thought to be hsTRPA1 mediated in US stimulated hsTRPA1 sonogenetic experiments.

Due to technical challenges of running pharmacology experiments in addition to US stimulation, we did not test TRPA1 inhibitors during whole-cell recordings in neurons. The most likely explanation for the currents observed in Cre-only neurons would be voltage gated sodium channels (Tyler et al 2008), though we did not test this experimentally. We agree with the reviewer's comment regarding the release of signaling molecules that may have down-stream effects, such as modulation of plasticity by activity dependent mechanisms. Although very interesting, a deeper investigation into the TRPA1 dependent plasticity, or the intermediary mechanisms elicited by US was out of the scope for this manuscript.

What is the number of events in experiments shown in Fig. 4g,h. This could be as relevant as amplitude (Fig. 4i) to report? Please define time frame of these experiments and those in Fig. 4j,k. Also, define pulse duration. Could action potentials be repeated?

Number of experiments, transfections, preparations, animals etc., is unclear here and in most experiments/figure legends throughout the study.

Details on cell number and pulse duration included in figure legend (Fig.4). Responded to multiple stimulations during whole-cell recordings in line 322.

Extended Data Fig. S9; no significance is shown in e.

Included in revised legend for Fig.S9.

hsTRPA1 confers ultrasound sensitivity in vivo

Fig. 5; statistical analysis in legend? What is the number of animals used?

Fixed.

Have the authors considered to express the hsTRPA1/N-tip d.r. chimera as a negative control (as well as in primary neurons)?

We agree that the hsTRPA1/N-tip deletion is a good control, but this is beyond the scope of this study. We are currently working on a follow up manuscript probing additional details of the mechanism and will include this experiment in that study.

Is it possible that hsTRPA1 is activated by free radicals formed by US stimulation in vivo as well as in vitro, and that species differences observed in the present study are influenced by TRPA1 channel species-dependent redox sensitivity?

Data in S5d

DISCUSSION

Line 358; hTRPA1 is intrinsically mechanosensitive (Moparthy and Zygmunt Cell Calcium 2020) –

Added to the discussion

Line 372; Mosquito and hTRPA1 are thermo- and mechano-sensitive also without the N-ARD (Survery et al JBC 2016; Moparthy et al PNAS 2014; Moparthy and Zygmunt Cell Calcium 2020).

Added to the discussion

Line 382; MCD reduced both sensitivity and Emax responses to AITC in mTRPA1.

Added to results

Reviewers' Comments:

Reviewer #1:

Remarks to the Author:

The authors have addressed all of the questions.

I recommend to accept the revised manuscript.

Reviewer #2:

Remarks to the Author:

The authors have addressed my comments. This is a work well done.

Reviewer #3:

Remarks to the Author:

The authors have satisfactorily addressed my concerns. However, please check Extended Data Fig. S2 and legend where a, b, c to l is not correct. Please check HC030031 (not e.g., HC03031) throughout the manuscript and elsewhere including figures. Also, update ref #42.

Response (in black) to the reviewer's comments (in blue)

Reviewer #1 (Remarks to the Author):

The authors have addressed all of the questions.
I recommend to accept the revised manuscript.

We thank the reviewer for their efforts.

Reviewer #2 (Remarks to the Author):

The authors have addressed my comments. This is a work well done.

We thank the reviewer for their efforts.

Reviewer #3 (Remarks to the Author):

The authors have satisfactorily addressed my concerns. However, please check Extended Data Fig. S2 and legend where a, b, c to l is not correct. Please check HC030031 (not e.g., HC03031) throughout the manuscript and elsewhere including figures. Also, update ref #42.

We have fixed Extended Data Fig 2 and its legend. Also, we edited our manuscript to use HC-030031 throughout and updated reference 42, which is now #34.